# Genomic signatures of migratory preference and historical whaling in eastern South Pacific humpback whales
Enrique Celemín[1,2,3], Jorge Acevedo[2], Linda Hagberg [4,5,6], Cristina Castro [7], Juliana Castrillón[7], Pedro Valenzuela [2], Luis A. Pastene[2,8] & Ralph Tiedemann [1,2] ✉

Understanding the genetic consequences of migratory behavior and drastic population contractions is essential for the conservation of baleen whales. Here, we investigated population structure and demographic history of eastern South Pacific humpback whales using whole genomes across two feeding and one breeding ground. Nuclear genomic analyses revealed no clear population structure between feeding grounds, suggesting high gene flow and panmixia, despite divergent migratory destinations. However, mitogenomic data showed strong clustering of the different feeding grounds, likely reflecting strong female-driven transgenerational tradition in seasonal migration. Demographic reconstructions identified a population expansion after the Last Glacial Maximum and a pronounced recent population decline consistent with industrial unsustainable whaling in the early 20th century. Despite this bottleneck, current genome-wide diversity remains substantial, possibly reflecting the buffering effect of humpback's long generation time. Nevertheless, the full genetic consequences of this demographic contraction may not yet be apparent and could become evident in subsequent generations.

Humpback whales (*Megaptera novaeangliae*), hereafter referred to as HWs, are a globally distributed baleen whale species that undertakes long-distance seasonal migrations that can reach distances of ~8500 km[1–4]. In the Southern Hemisphere, they migrate between high-latitude regions where they typically feed during the austral summer/autumn, and low-latitude regions where they breed and calve during the austral winter/spring[5,6]. Eastern South Pacific (ESP) HWs are one of the seven breeding populations or stocks (G in this case) recognized in the Southern Hemisphere by the International Whaling Commission (IWC)[7]. This population breeds and calves in a continuous range from northern Peru (4°S) to southern Nicaragua ( ~ 11°N) in the eastern South and Central Pacific Ocean[3,8–11] and migrates southerly to the western coast of the Antarctic Peninsula in the Southern Ocean[1,11–13]. Furthermore, a small fraction of the HWs also uses the waters of southern Chile, as the Corcovado Gulf and the Magellan Strait as feeding grounds[14–16].

Different lines of evidence suggest some degree of differentiation between the HWs that feed at the Magellan Strait with those that feed in the more southern regions close to the Antarctic Peninsula. Photo-identification and proportion of white and black fluke coloration analysis suggests the existence of a latitudinal preference within the grounds: with Magellan Strait HWs being more likely to migrate to the northern range of the breeding ground (e.g., off Panama) whereas whales from the Antarctic Peninsula more likely to migrate to the southern breeding range (e.g., off northern Peru, Ecuador and Colombia)[11]. In addition, the population dynamics of the Magellan Strait HWs show lower annual growth rate (2.1%) and annual survival rates (0.892[17]), compared with the Antarctic Peninsula HWs (annual growth rate: 6.3%; annual survival: 0.929[18]). Moreover, previous genetic studies have shown a pronounced differentiation in mitochondrial control region haplotypes between the Magellan Strait HWs and those from the Antarctic Peninsula feeding ground[19,20]. These particular characteristics could satisfy the preliminary criteria for the recognition of distinct "management units"[21] or "distinct population segments"[22]. While these mitochondrial-based studies suggest potential population subdivision, they represent only the maternal lineage. However, to date, very little is

[1]University of Potsdam, Institute of Biochemistry and Biology, Unit of Evolutionary Biology/Systematic Zoology, Potsdam, Germany. [2]Centro de Estudios del Cuaternario de Fuego Patagonia y Antártica (CEQUA), Punta Arenas, Chile. [3]Department of Integrative Biology, University of South Florida, Tampa, FL, USA. [4]Department of Evolutionary Genetics, Leibniz Institute for Zoo and Wildlife Research (IZW), Berlin, Germany. [5]Berlin Center for Genomics in Biodiversity Research (BeGenDiv), Berlin, Germany. [6]Museum für Naturkunde, Leibniz Institute for Evolution and Biodiversity, Center for Integrative Biodiversity Discovery, Berlin, Germany. [7]Pacific Whale Foundation-Ecuador, Puerto López, Ecuador. [8]Institute of Cetacean Research, Tokyo, Japan.
✉e-mail: tiedeman@uni-potsdam.de

known about the nuclear genetic structure of ESP-HWs, and no assessment of population structure using whole-genome sequencing data has been conducted, limiting both population dynamics understanding and effective conservation planning for the ESP-HWs.

In the Southern Hemisphere, humpback whales were heavily exploited by modern commercial whaling, which began in 1903 in the Magellan Strait waters[23], and rapidly expanded throughout the eastern South Pacific. Although the IWC banned humpback hunting in 1963, some operations continued until 1975 when the last factory in northern Peru closed[24]. This prolonged exploitation severely depleted all seven southern hemisphere stocks by the mid-20th century, although most began recovering in the 1960s–1970s[25]. The extent of this recovery (measured as a proportion of pre-exploitation abundance in 1900) ranged from 0.13 to 0.97 across different stocks and sub-stocks[25,26]. The IWC estimated the total abundance of southern humpback whales around 138,000 (95% probability intervals: 111,900–198,000) in 1900, with an estimated abundance in 2015 of 97,000 (78,000-117,500), implying a recovery level of 0.70[26]. Regarding the stock relevant for this study (Stock G), its population size decreased drastically from before 1920, remained low between 1920 and 1960, and started to increase from the 1960's[27]. The population abundance in 1900 was estimated at 11,600 (10,600–14,900) HWs, and that estimated in 2015 was 9700 (8500–10,200), implying a recovery level of 0.93 (0.74–0.98)[26]. Recent photo-identification mark-recapture analyses have estimated the population size of ESP-HWs at about 11,800 individuals, representing 1.6% and 18.1% higher than to the estimates of 1900 and 2015, respectively, and an annual growth rate of 5.1%[28].

This documented population contraction lasted for approximately 70 years ( ~ three generations), and may have led to long-lasting negative effects on the genetic diversity of ESP-HWs. Populations with reduced effective population sizes ($Ne$) will experience increased genetic drift and eventually be at risk for inbreeding depression, consequently affecting fitness and limiting population growth[29]. However, estimating the impact of the whaling bottleneck on the genetic diversity of baleen whale species has long been challenging due to a combination of limited genetic data and the life history traits (i.e. long life span and generation time) of these species[30]. Traditional genetic markers capture long-term effective sizes rather than recent demographic events[31], which has hindered the ability to directly detect recent population contractions. Recent studies have begun to leverage whole-genome sequence data to infer historical population sizes, bottlenecks, and recovery patterns in fin[30,32], bowhead[33,34], North Atlantic and Southern right[35], and blue whales[36]. Despite these advances, no genomic studies have yet addressed the genetic consequences of unsustainable whaling in humpback whales, an important gap in understanding their current genetic health and resilience.

Here, we used whole-genome sequencing data to investigate the genomic signatures of migratory preference between feeding grounds and to assess the impact of industrial whaling on ESP-HWs. Specifically, we aimed to: (1) determine whether substructure exists among feeding grounds using both nuclear and mitochondrial data, (2) reconstruct the historical and recent demographic history of ESP-HWs to measure the potential population bottleneck caused by industrial whaling and (3) evaluate whether industrial whaling negatively affected current genomic diversity levels of ESP-HWs. Our findings shed light on the genetic structure patterns of ESP-HWs and reveal the genomic consequences of the whaling period and distinct migratory patterns in an illustrative baleen species.

## Results

To unravel population structure, describe the genomic diversity and to disentangle the evolutionary history of eastern South Pacific humpback whales (ESP-HWs), we re-sequenced the genome of 26 specimens, including individuals from one breeding ground (Ecuador, ECU) and two feeding grounds (Magellan Strait, MAG and Antarctic Peninsula, ANT)

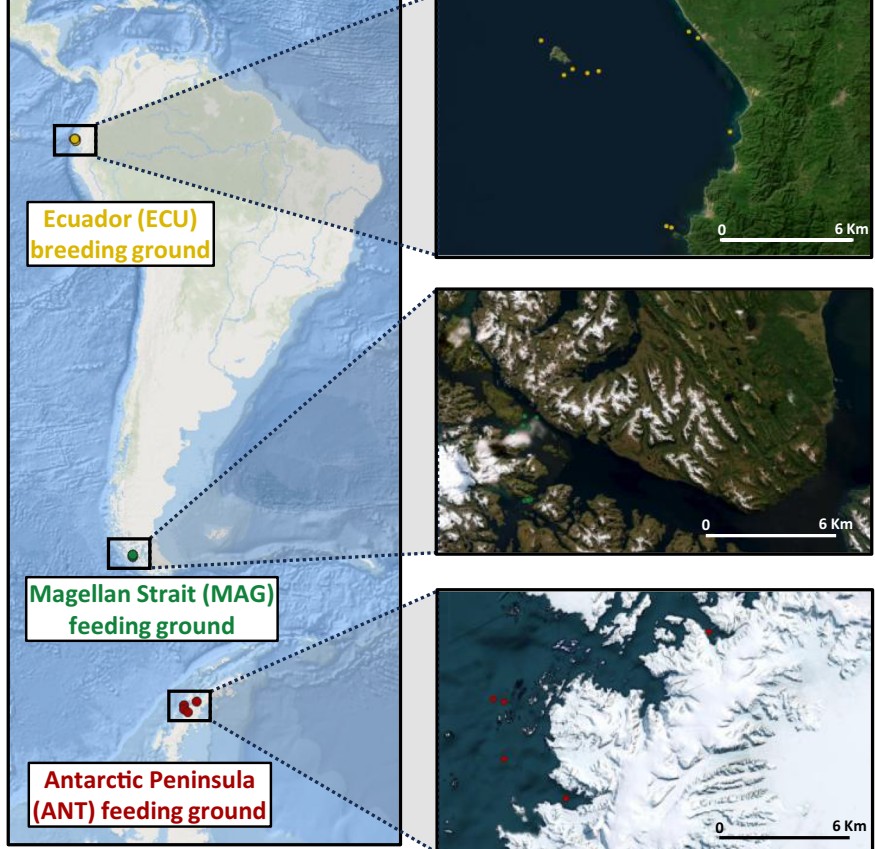

**Fig. 1 | Map of the sampling locations colored according to origin.** Ecuador breeding ground in yellow, Magellan Strait feeding ground in green and Antarctic Peninsula feeding ground in red.

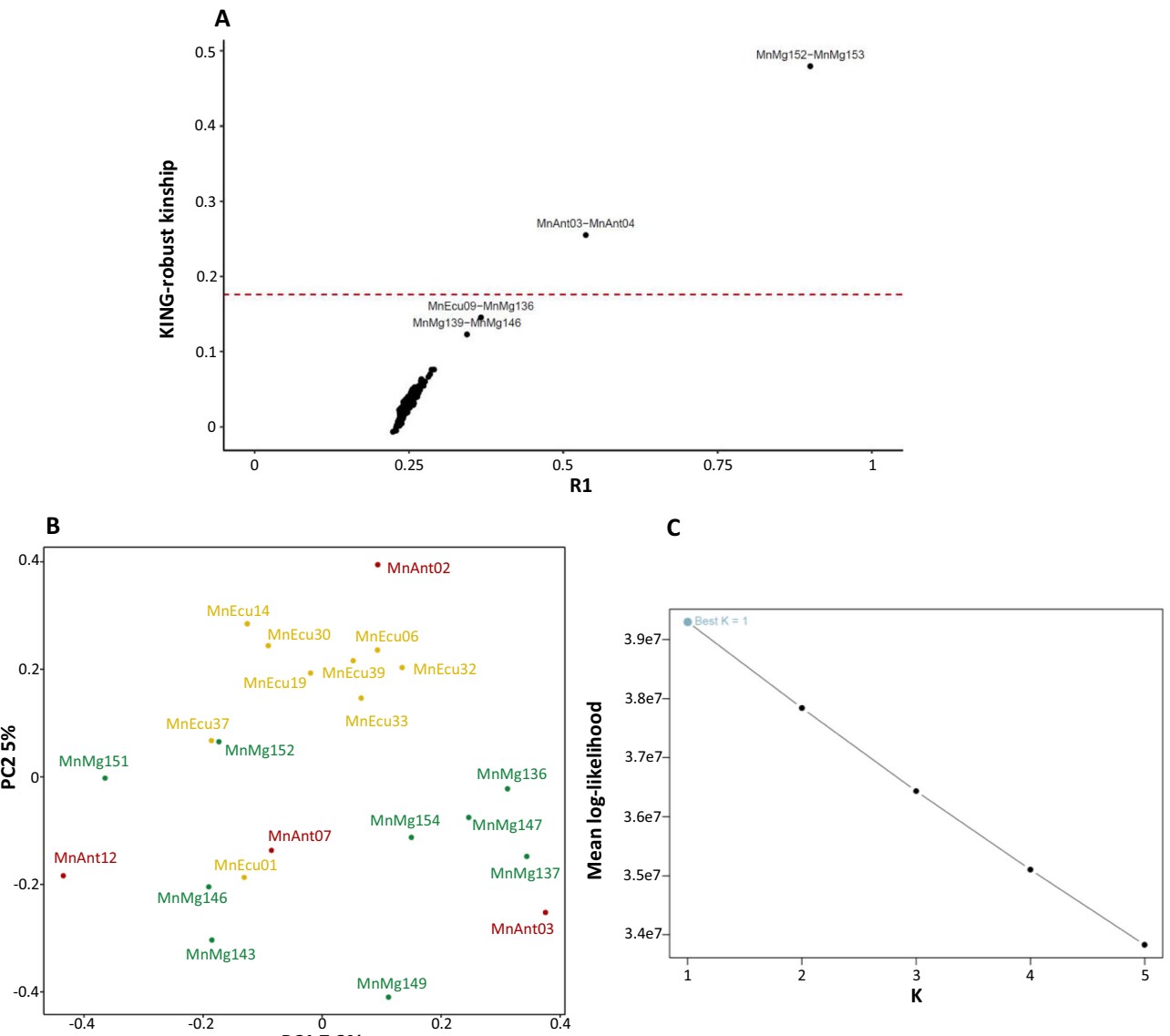

**Fig. 2 | Relatedness and population structure analysis of eastern South Pacific humpback whales. A** Relatedness analysis showing R1 and KING statistics for all individual pairs. The dash line represents the threshold of KING statistic for first-degree relatives ( ~ 0.177). (**B**) Principal component analysis (PCA) of the dataset without first and second-degree relatives (N = 22) showing the first and second PCs.

The coloring corresponds to different sampling locations: Ecuador breeding ground in yellow, Magellan Strait feeding ground in green and Antarctic Peninsula feeding ground in red. **C** Mean log-likelihood of the NGSAdmix runs plotted against the number of genetic clusters (*K*).

(Fig. 1). Sample biological information, sequencing, filtering, and mapping statistics are provided in Supplementary Table 1, while the different bioinformatic steps and analysis carried out are shown in Supplementary Fig. 1. After downsampling the individual genomes to a coverage of 5X, we identified 5,510,405 high-quality SNPs (MAF ≥ 0.05), of which 2,582,223 were unlinked (Supplementary Fig. 1). Relatedness analysis (Fig. 2A) identified two pairs of first-degree relatives and two pairs of second-degree relatives. Both pairs of first-degree relatives were sampled on the same day at proximate or identical locations, once in the Magellan Strait and once at the Antarctic Peninsula (Supplementary Table 1). Based on this, we filtered first and second-degree relatives to generate our final dataset of 22 unrelated HWs.

### Genetic structure

The population structure analysis, *PCAngsd* and *NGSAdmix* (Fig. 2), based on ~2.5 million unlinked SNPs, indicated that ESP-HWs form a single genetic population. The PCA (Fig. 2B) revealed no clear clustering by

feeding grounds, suggesting interbreeding across HWs seasonally migrating to these geographically separate feeding areas. Admixture analysis also supported this result as we identified *K* = 1 as the most likely (Fig. 2C; Supplementary Fig. 2; 3). The admixture analysis assuming *K* = 2 assigned roughly half of the individuals to one of two genetic clusters, but these clusters showed no clear geographic pattern. When *K* was set to 3–5, the results showed high levels of admixture and lacked any clear clustering pattern, further suggesting that these values of *K* were not appropriate (Fig. 2C). Pairwise genetic differentiation among regional grounds were uniformly low (0–0.002), indicating negligible genetic differentiation among sampling locations (Supplementary Table 2).

In contrast, both the mitochondrial genome haplotype network (Fig. 3) and phylogenetic tree (Supplementary Fig. 4) revealed some degree of geographic sub-structure. Haplotypes present in the Magellan Strait feeding ground showed close clustering of haplotypes. This monophyletic cluster (mito-cluster 1) included the most common haplotype (H6, see Supplementary Fig. 5), found across all three grounds, and ten of eleven haplotypes

**Fig. 3 | Mitochondrial haplotype median-joining network of eastern South Pacific humpback whales.** Ticks along the branch lengths denote nucleotide differences between the haplotypes found in each sample. The coloring corresponds to different sampling locations: Ecuador breeding ground in yellow, Magellan strait feeding ground in green and Antarctic Peninsula feeding ground in red.

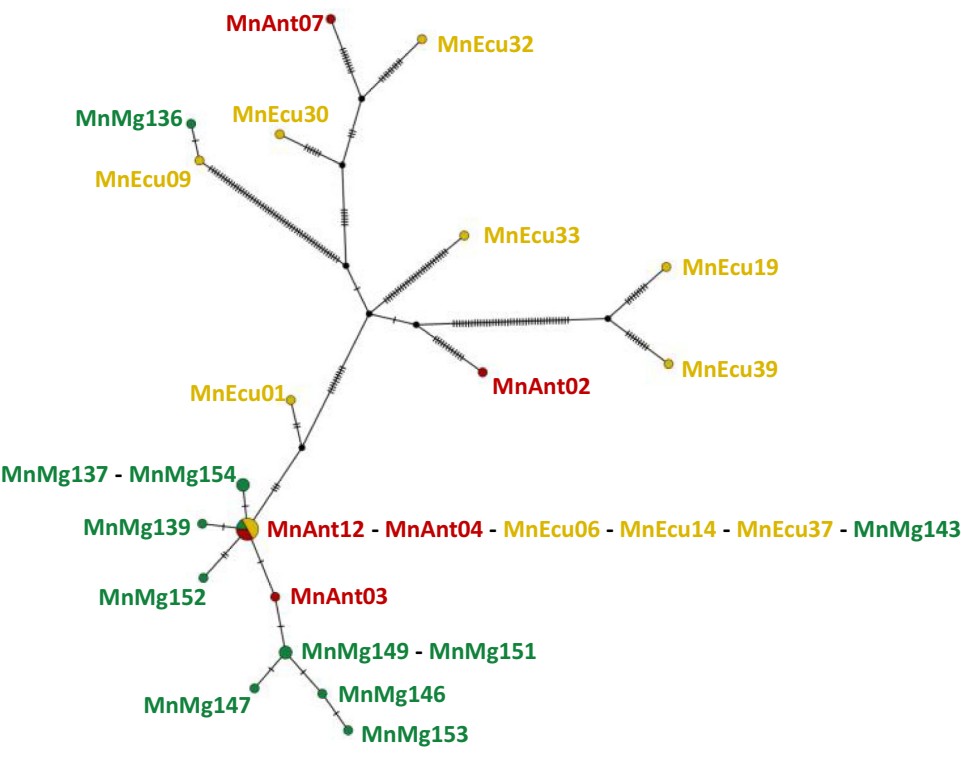

from the Magellan Strait, which exhibited a star-like pattern with only one or two mutations separating haplotypes. The remaining haplotypes were more divergent from one another (forming several only partly resolved lineages/clusters in the phylogenetic tree, Supplementary Fig. 4). Among them were most haplotypes present in the Ecuador and Antarctic Peninsula grounds, plus one haplotype from the Magellan Strait (MnMg136). No mito-cluster/lineage exclusively contained individuals from a single region, nor did any cluster/lineage encompass all individuals from that region (Supplementary Fig. 5).

## Demographic history

Our demographic analysis in $\delta a \delta i$ indicated that the 2-epoch model best fits our data (Supplementary Fig. 6; Supplementary Table 3). The 1-epoch model can be rejected in favor of the 2-epoch model, but the 2-epoch model cannot be rejected in favor of the 3-epoch model after a LRT (Supplementary Table 4). The 4-epoch model did not converge (Supplementary Table 3) suggesting overfitting, and was therefore excluded. Both the 2-epoch and 3-epoch models suggested a population expansion beginning approximately 53,000 years ago (ya; ~2485 generations ago), from an ancestral effective population size ($Ne$) of ~3700 to a $Ne$ of ~394,000 (Fig. 4A, B; see Supplementary Table 5 for model-specific estimates and 95% confidence intervals). Additionally, the 3-epoch model revealed a sharp recent decline in population size (from ~394,000 to ~2500) in the present generation (0 generations ago) (Fig. 4B). Fixing the onset of the 3rd epoch to higher values (i.e., 1, 5, 10, 20, or 30 generations) yielded lower likelihoods (Supplementary Table 3).

The historical demographic reconstruction using $SMC++$ indicated a population increase from ~100,000 to ~25,000 ya, with $Ne$ rising from ~10,000 to ~26,000, followed by a steady decline to contemporary $Ne$ levels lower than 500 (Fig. 4C). The $Ne$ trajectories of the HW of the different grounds exhibited a similar pattern, highlighting shared ancestry and interbreeding among grounds (Supplementary Fig. 7).

Estimates of recent $Ne$ changes from the software $GONE$ revealed a sharp decline beginning ~750 ya ( ~ 30 generations ago), reaching the lowest $Ne$ ( ~ 1200) around 175 ya ( ~ 8 generations ago) (Fig. 4D). While the $Ne$ trajectories of MAG and ECU HWs displayed similar patterns, the ANT

HWs show a notable $Ne$ increase after ~1000 ya (50 generations ago), suggesting low concordance among $GONE$ runs, likely due to the small sample size ($N = 4$) (Supplementary Fig. 8).

## Genome-wide patterns of variation and runs of homozygosity

Using genotype likelihoods, we estimated individual genome-wide heterozygosity (heterozygous sites/total sites) from the site frequency spectrum. Estimates ranged from 0.0013 to 0.0015, with ANOVA showing no significant differences between sampling grounds (Fig. 5A; $F_{2,23} = 0.064$, $p = 0.938$). Similarly, no significant differences were found in the proportion of the genome in runs of homozygosity (FROH) among grounds (Supplementary Table 6 and Fig. 5B; ANOVA: $F_{2,23} = 2.245$, $p = 0.129$). The majority of runs of homozygosity (ROHs) were short (100–300 Kb; Fig. 5D; Supplementary Table 6), with mean ROH length ( ~ 172 Kb) consistent across grounds (ECU = 175.4 Kb; MAG = 170.3 Kb; ANT = 171 Kb).

## Discussion

Our nuclear genomic results suggest that eastern South Pacific humpback whales (ESP-HWs) are largely panmictic, with no clear genetic structure distinguishing individuals from the Magellan Strait (MAG) and Antarctic Peninsula (ANT) feeding grounds. PCA (Fig. 2B), admixture (Supplementary Fig. 2) and Fst (Supplementary Table 2) analyses consistently revealed no clustering by feeding region, suggesting interbreeding regardless of migratory destination. Demographic reconstructions using $SMC++$ further support this view, as the effective population size ($Ne$) trajectories of the three different grounds are remarkably consistent, pointing to a long history of connectivity (Supplementary Fig. 7). However, individuals from the Magellan Strait exhibit a tendency towards lower heterozygosity than those from other regions, with limited variance among individuals aside from a single outlier (Fig. 5A).

The mitogenome (mtDNA) analysis reveals a more complex and regionally differentiated picture compared to nuclear data (Supplementary Fig. 5). The haplotype network constructed from whole mitogenomes (Fig. 3) displays a striking star-like clustering of nearly all haplotypes sampled in the Magellan Strait. Such a pattern is commonly associated with population expansion and/or founder effects[37,38] and hence fits well to the

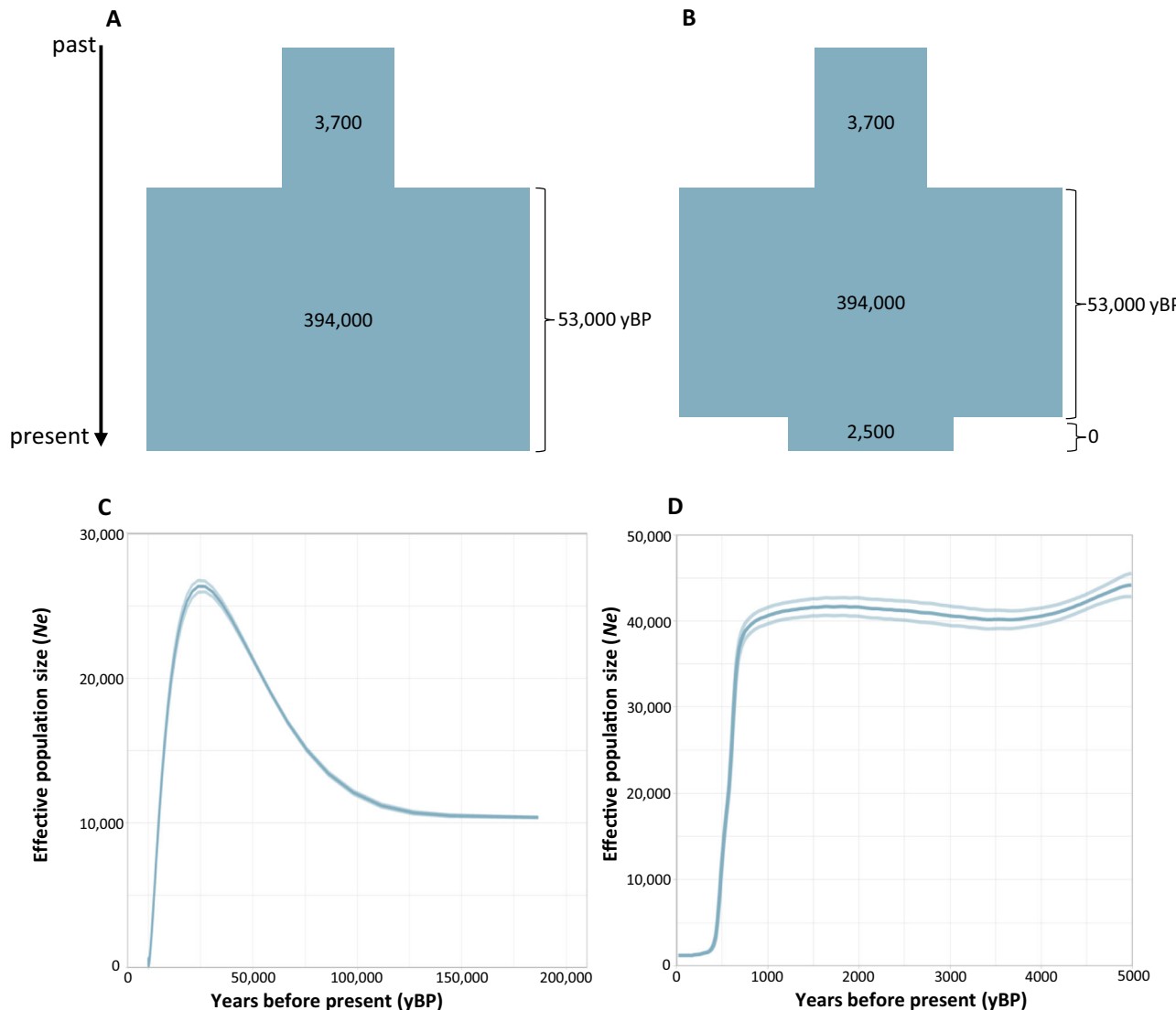

**Fig. 4 | Demographic history of eastern South Pacific humpback whales.**
**A** 2-epoch model with a population expansion during the last glacial maximum
(LGM). **B** 3-epoch model representing the same population expansion during the
LGM and a population contraction in the very recent time, likely due to historical
whaling ( ~ 1900–1975). **C** Changes in historical effective population size (*Ne*)
through time inferred with the coalescence method applied in *SMC + +*. **D** Changes

on recent effective population size (*Ne*) through time inferred with the linkage
disequilibrium approach implemented in *GONE*. Solid *Ne* curves depict the geo-
metric mean over 100 independent estimates, while blurred lines represent the 95%
confidence intervals. The X-axis represents time before present in years while the
Y-axis depicts *Ne*. Note the different time scales among the applied approaches.
Inferences are based on *n* = 22.

postglacial expansion inferred from our nuclear analyses. Interestingly, the
central haplotype of this cluster is found in all regions, while the derived
haplotypes are mostly restricted to the Magellan Strait. This could indicate a
particularly successful HW matriline from which HWs preferentially
migrate to the Magellan Strait. However, this structuring is not absolute.
One individual from MAG harbors a haplotype markedly divergent from
the others, separated by numerous substitutions. This outlier may reflect the
presence of an additional maternal lineage or ongoing movement between
breeding (and associated feeding) grounds.

We identified two few first-degree relatives (i.e. only two pairs) for any
robust kinship-based population structure inference. In both occasions, the
two relatives were sampled in close proximity on feeding grounds, providing
anecdotal evidence for some kin-related migration preferences. These
findings suggest that mtDNA lineage distribution may be shaped by
migratory preferences. As for the Magellan Strait, these migratory traditions
may be maintained across generations, creating a pattern of distinct, yet
closely related mitochondrial haplotypes found there. However, such pre-
ferences do not prevent nuclear genetic exchange at the population level.

The discordance between nuclear and mitochondrial signals is not
unexpected, given their different modes of inheritance. Unlike nuclear
DNA, which is inherited biparentally and recombines, mtDNA is mater-
nally inherited and more prone to genetic drift, often revealing population
structure over shorter timescales[39,40]. In cetaceans, females tend to exhibit
greater philopatry while males are more dispersive[40], an asymmetry that can
obscure population structure in nuclear markers but leave detectable
imprints in mtDNA[41]. This is particularly relevant for humpback whales,
where females typically maintain stable migratory routes[40], making the
mitogenome a sensitive indicator of localized demographic and evolu-
tionary histories.

Detecting population bottlenecks caused by unsustainable industrial
whaling using genetic diversity estimates from contemporary samples is
challenging, since the influence of changes in population size on genetic
diversity is slow, relative to the temporal scale of human-induced events[42,43].
Moreover, the extent of genetic variation loss depends not only on the
severity of the bottleneck, but also on its duration relative to the life-span and
generation time of the affected species[30]. In the case of eastern South Pacific

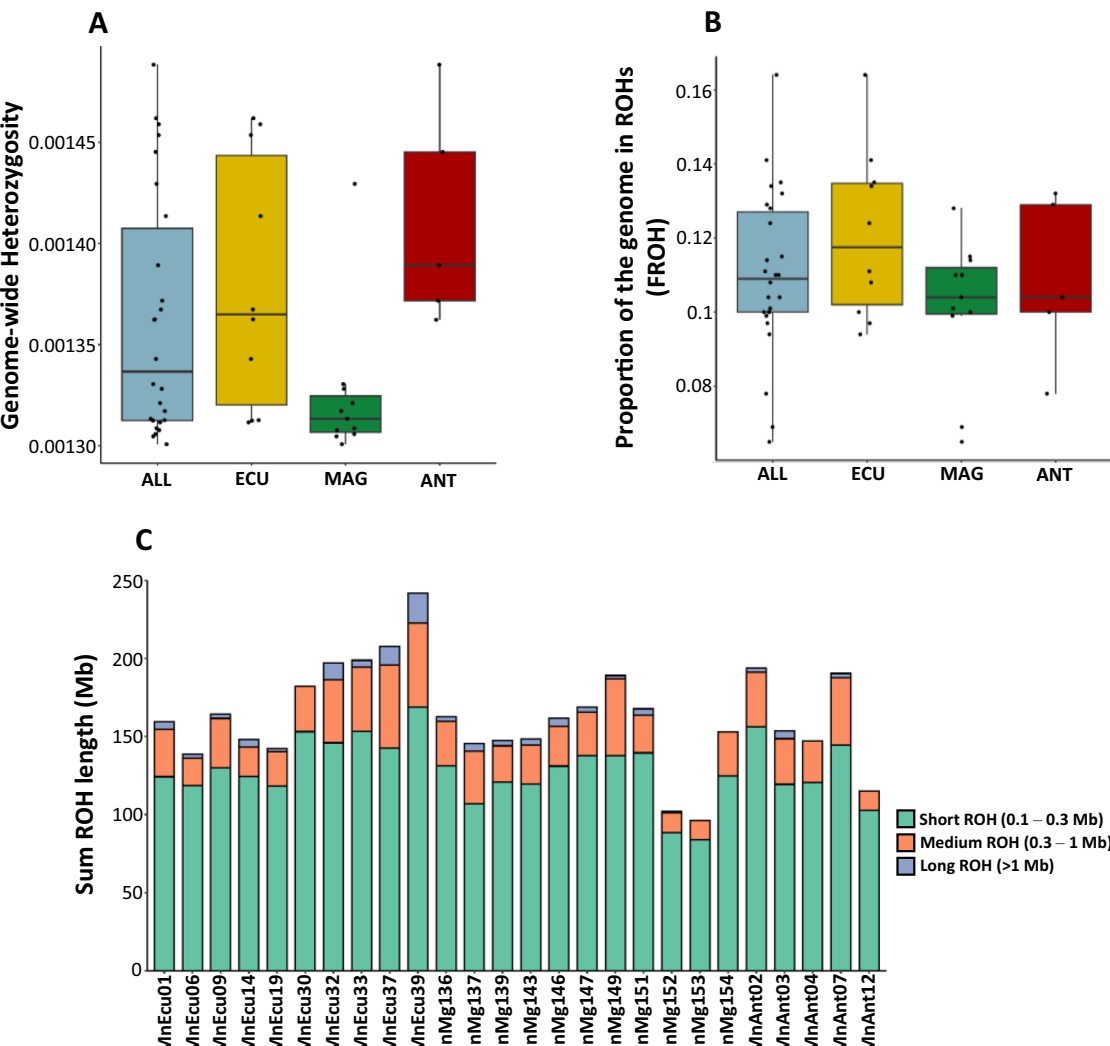

**Fig. 5 | Population genomics summary statistics of eastern South Pacific humpback whales. A** Genome-wide heterozygosity and (**B**) inbreeding coefficient (FROH) for all eastern South Pacific humpback whales (light blue) and for each sampling location separately (Ecuador breeding ground in yellow, n = 9; Magellan Strait feeding ground in green, n = 9; and Antarctic Peninsula feeding ground in red,

n = 4). Box plots show the median (center line), interquartile range (IQR; box limits), and whiskers extending to the most extreme values within 1.5× IQR; all individual data points are shown as jittered dots. **C** Barplot showing summed lengths of short (0.1–0.3 Mb), medium (0.3–1 Mb) and long ( > 1 Mb) ROH per individual.

humpback whales (ESP-HWs), the whaling bottleneck occurred roughly 100 ya and lasted about 40 years[26,27], whereas individuals can live up to 95 years[5] and have a generation time of approximately 21.5 years[44].

To better understand the genomic impact of whaling, we applied three complementary methods ($\delta a \delta i$, $SMC + +$, and $GONE$) to reconstruct the demographic history of ESP-HWs (Fig. 4). These methods offer complementary advantages: $\delta a \delta i$ provides robust inference of recent and intermediate demographic changes through its reliance on the site frequency spectrum (SFS)[45], $SMC + +$ uses Sequential Markov Coalescent methods to capture fine-scale historical demographic trajectories[46], and $GONE$ offers direct detection of contemporary effective population size through linkage disequilibrium patterns[47]. Both $\delta a \delta i$ and $SMC + +$ revealed a prolonged period of population expansion of ESP-HWs following the Last Glacial Maximum, beginning approximately 53,000 ya. This expansion is consistent with favorable post-glacial environmental conditions that likely facilitated population growth and range expansion[48]. Both the 2-epoch and 3-epoch models fit the data well (Supplementary Fig. 6, Supplementary Table 3). However, our ability to detect parameter rich scenarios (i.e., a more than 2-epoch scenario) may have been limited, as the lack of an annotated reference genome prevented us from designating neutral areas of the

genome, potentially affecting statistical power. As the 2-epoch and the 3-epoch scenarios yielded very similar likelihoods (cf. Supplementary Table 3), we also evaluated the results of the 3-epoch model. Indeed, the parameter estimates of the 3-epoch model reflect a recent sharp decline in effective population size ($Ne$), as also inferred by both $SMC + +$ and $GONE$. Specifically, $SMC + +$ detected a gradual decline in $Ne$ following the post-glacial expansion, whereas $GONE$ identified a marked reduction beginning around 750 years ago (ya). Together, these findings suggest that the demographic history of ESP-HWs consists of an ancient population expansion and a recent population contraction.

The onset of the final epoch in the 3-epoch $\delta a \delta i$ model is estimated at generation 0 (i.e., the present generation). Fixing the onset of the final epoch to higher values preceeding the time of commercial whaling (i.e., 10, 20, 30 generations ago) yields lower likelihoods (Supplementary Table 3). The inferred decline in the 3$^{rd}$ epoch hence potentially reflects the persistence of genetic signals from the whaling-era bottleneck, which unfolded within the last five ESP-HW generations. Indeed, given HWs' longevity (up to 95 years) and their long generation time ( ~ 21.5 years), individuals sampled in this study may have been born during or shortly after the bottleneck or are direct descendants of survivors from that period, which might influence very

recent parameters estimation. Likewise, the timing of the decline inferred by *GONE* ( ~ 750 ya) does not precisely coincide with historical records of peak whaling of ESP-HWs[26,27]. Once again, the accuracy of these estimates may be limited by methodological constraints, i.e. *GONE* requires a species-specific recombination map, which is not currently available for humpback whales. We used a recombination rate typical of large mammals (an approach previously applied in cetacean studies[29]) as well as a fixed generation time of 21.5 years, which may not fully capture life history variation. Furthermore, because *GONE* estimates *Ne* in discrete blocks of generations[47], the first four generations are necessarily identical (see the flat *Ne* trajectory in the initial generations of Fig. 4D), which complicated the detection of the whaling bottleneck. Discrepancies in time estimates are not uncommon in demographic inferences based on genome data; similar mismatches have been reported in other whole-genome studies of baleen whales[32,34,35]. These issues underscore the challenges of using genetic data to precisely time recent demographic events. Future studies that incorporate temporally stratified or epigenetically aged samples, along with an annotated reference genome, and genomes sequenced at a higher coverage may help refine these estimates and provide more accurate demographic reconstructions[30,32].

Nonetheless, all of our demographic approaches consistently indicated a pattern of historical population expansion followed by a pronounced recent decline in the ESP-HW population, with *δaδi* and *SMC++* capturing the longer-term demographic trajectory spanning millennia, while *GONE* provide higher-resolution detection of the sharp, recent collapse in effective population size coinciding with the industrial whaling era. Yet we urge caution when interpreting specific temporal estimates and population sizes due to methodological constraints. Other factors, such as climatic or anthropogenic influences may have contributed to this decline. In particular, the so-called Little Ice Age (LIA) in the medieval might have affected humpback whales. In the Southern hemisphere, LIA was apparent between about 1600 and 1700 ([49]; translating into about 20 and 15 humpback generations ago). In this context, our *δaδi* modeling argues against LIA as the major driver of the decline, as the 3-epoch simulation estimates the decline to happen in the current generation (generation 0), while forcing the model to implement an earlier decline results in decreased likelihoods (Supplementary Table Table 3). The most plausible explanation for the recent and pronounced reduction in effective population size hence remains the well-documented period of unsustainable industrial whaling (1903–1975).

Despite genomic evidence for a pronounced recent population decline of ESP-HWs, genome-wide heterozygosity remains relatively high (0.0013–0.0015; Fig. 5A). These values are consistent with those reported for other baleen whales as the fin, blue and North Pacific Right whale[35,50]. In contrast, our heterozygosity estimates are substantially higher than those observed in critically endangered cetaceans such as the Vaquita, Baiji, and Māui dolphin, as well as in heavily exploited baleen species like the North Atlantic right whale[35,50]. Similarly, the proportion of the genome in runs of homozygosity (FROH) is relatively low (Fig. 5B). These patterns suggest that the whaling bottleneck may have been brief or followed by rapid recovery, allowing the population to retain much of its genetic diversity. The distribution of runs of homozygosity (ROH) supports this interpretation: most ROHs were short (100–300 Kb; Fig. 5D), with comparable mean lengths across regions, consistent with an ancient rather than recent bottleneck. Alternatively, the full genetic consequences of the recent bottleneck may not yet have manifested due to the species' long generation time, and further erosion of diversity may still be expected in future generations, which highlights the need for forward-time simulations to explore these possibilities[29,30,51]. Furthermore, we acknowledge that our coverage is only ~5X after downsampling. This is a common coverage in studies on non-model organisms where researchers have to balance coverage, sample sizes, and associated costs (see e.g.[51,52] for comparable genomic studies on small cetaceans). Indeed, we strictly followed recommendations for population genomics on low-coverage data, most notably the use of likelihood-based methods[53]. Nonetheless, we cannot exclude that low coverage and the use of a reference genome from a different population (Hawaii, North Pacific) may

have led to an overestimation of genome-wide heterozygosity and under-estimation of ROH lengths[54].

Overall, our findings indicate that while migratory preferences in ESP-HWs may shape the distribution of maternal lineages, they do not currently limit gene flow at the nuclear level or lead to distinct population structure. The discordance between mitochondrial and nuclear signals underscores the importance of incorporating multiple genetic markers to uncover sex-biased dispersal and subtle structuring that might otherwise go undetected[55]. Notably, the elevated recapture rates and site fidelity in the Magellan Strait[56] correspond to our distinct mitochondrial clustering patterns, indicating interconnected demographic and genetic structure. For a better understanding of the origin of the HWs feeding in the Magellan Strait, genomic analyses of HWs from more northern breeding grounds (e.g., off northern Colombia, Panama and Costa Rica) would be desirable.

Our demographic analyses suggest that ESP-HWs experienced a post-glacial population expansion followed by a substantial recent contraction, presumably driven by unsustainable industrial whaling in the 20th century. Despite this sharp reduction in effective population size, the long-term genetic consequences appear moderate. Given that genetic diversity declines over time as a function of the number of generations since a bottleneck, the modest reduction observed likely reflects the long generation time of humpback whales ( ~ 21.5 years), the relatively short duration of intensive whaling ( ~ 70 years), and potential signs of demographic recovery following the 1963 whaling moratorium for this species. However, the delayed genetic effects of a bottleneck in long-lived species may not yet be fully apparent. Our results align with contemporary demographic data showing that ESP-HWs have recovered to approximately 11,784 individuals, yet the annual growth rate of 5.1% remains below of both the theoretical maximum (11.8%) and rates observed in other recovering humpback populations[28]. This demographic limitation may signal persistent genetic bottleneck effects not yet apparent in diversity measures, thus continued population surveys and genomic monitoring will be essential to assess the long-term genetic health and evolutionary potential of ESP-HWs.

## Methods
### Sampling and laboratory procedures

Surveys of HWs were conducted in two migratory habitats, the feeding grounds in the Magellan Strait (Chile) and Antarctic Peninsula during the austral summer season (February-April), and the breeding ground in Machalilla National Park waters off Ecuador during the winter austral season (August and September; see Supplementary Table 1 for details). Biopsy skin samples were collected from free-ranging whales that showed no visible signs of illness. In Antarctic Peninsula, the samples ($n = 5$) were collected using a Paxarm MK24C remote biopsy system[57]. The biopsies in Ecuador ($n = 10$) and Magellan Strait ($n = 11$) were collected using 7 mm diameter dart tips made of surgical steel (Ceta-dart), targeting the upper flank near the dorsal fin propelled by crossbows of 150 and 175 lb[58], respectively. All skin samples were preserved in 2 mL cryotubes containing 95% ethanol and stored at 4 °C until DNA extraction.

Genomic DNA was extracted from 26 biopsies at the Genetics and Genomics Laboratory of Fundación CEQUA. For each sample, 25 mg of skin tissue was incubated overnight at 56 °C with proteinase K and buffer ATL (Qiagen, Hilden, Germany). DNA lysis and purification were performed using the DNeasy Blood & Tissue Kit (Qiagen), following the manufacturer's protocol, with a final elution volume of 120 μl in AE buffer. DNA quality and quantity were rigorously evaluated using a Qubit Fluorometer and a NanoDrop ONE (Thermo Fisher Scientific) to ensure they met the requirements for whole-genome sequencing. A 30 μl aliquot of each sample was then sent to AZENTA Inc. (USA), where whole-genome libraries were constructed using the NEBNext Ultra II DNA Library Prep Kit for Illumina (New England Biolabs, Ipswich, MA, USA) following the manufacturer's recommendations. Briefly, genomic DNA was acoustically sheared using a Covaris LE220 instrument; fragmented DNA was cleaned up, end-repaired, and adenylated at the 3′ ends, then NEBNext adapters

were ligated and libraries enriched by limited-cycle PCR. Final libraries were validated on an Agilent TapeStation (Agilent Technologies, Palo Alto, CA, USA) and quantified using both Qubit 2.0 Fluorometer and by real-time PCR (KAPA Biosystems, Wilmington, MA, USA). Equimolar pools of indexed libraries were clustered onto an S4 flow cell of an Illumina NovaSeq 6000 instrument (Illumina, San Diego, CA, USA) and sequenced in a $2 \times 150$ bp paired-end configuration (targeting ~350 million paired reads per lane, ~105 GB output). Image analysis and base calling were performed by NovaSeq Control Software, and raw BCL files were converted to FASTQ and demultiplexed with Illumina bcl2fastq v2.20 software, allowing one index mismatch.

### Genomic data processing and SNP calling
First, low-quality ( < 15Q) and too short ( < 75 bp) raw reads were filtered with the software *Fastp* (v.0.23.2[59]). The remaining filtered reads were mapped to the humpback whale chromosome-level genome assembly (GCA_041834305.1[60]) using the *Bwa mem* algorithm (v.0.7.17[61]) with default settings, and *Picard* (v.2.27.2[62]) was used to add read groups (*AddOrReplaceReadGroups)* and to remove PCR and sequencing duplicates (*MarkDuplicates*). Thereafter, the mapped reads were realigned around indels with *Gatk* (v.3.8.1[63]). We used *Samtools* (v.1.19[64]) to remove 1,070,425,330 bp (34.02% of the genome) embedded in repetitive regions (previously identified with *RepeatMasker* (v.4.1.5[65]) and the *dfam 3.7*[66] database) and the sex chromosomes from the mapped reads. Additionally, we filtered reads with mapping quality below 30 and regions with insufficient (1/3 mean coverage) and excessive (2x mean coverage) depth, previously estimated with *Angsd* (v.13.2.0[67]). Finally, to have a consistent coverage among samples we downsampled our data to the minimum coverage presented by any sample (5X) using *Picard DownsampleSam.*

We called SNPs by calculating genotype likelihoods with *Angsd*, using the *Samtools* model (GL 1), keeping SNPs with a minimum minor allele frequency (MAF) of 0.05, having data in a minimum 75% of the individuals and a SNP $p$ value $< 1e^{-6}$. Since genotype likelihoods take into account genotype uncertainty and allow to obtain reliable SNPs at low coverages[53], we decided to use them in all the analyses that did not require hard genotype calls (relatedness statistics, population structure and heterozygosity estimation). Since related individuals might influence genetic structure patterns and analysis at the population level, we calculated relatedness statistics using the genotype likelihoods with the software *NgsRelate* (v.2.1[68]) to identify pairs of first-degree relatives. Specifically, we used both KING and R1 statistics because they complement each other: KING is robust to population structure for inferences of first-degree relationships, while R1 provides a more sensitive estimate of pairwise relatedness across the full range of relationships.

For the analyses that required genotype calls (Fst, ROH and demographic history analysis), we converted the hard calls generated in *Angsd* (.geno file) into a .vcf file using a Python script (*genoToVCF.py*[69]). Then, we filtered indels (--remove-indels), singletons (--max-mac 2) and retained only biallelic sites (--max-alleles 2) present in all individuals (--max-missing 1) using *Vcftools* (v.0.1.16[70]).

To obtain the mitogenomes from our whole-genome re-sequencing reads, we repeated the bioinformatic pipeline using the mitogenome reference of the humpback whale (MF409246.1[71]). Then, from the mapped reads we called consensus genotypes by outputting the most frequently observed genotype (-doFasta 2) using *Angsd*.

### Population structure analysis
We studied the genetic structure of ESP-HWs by performing PCAs and admixture analyses on a set of unlinked genotype likelihoods. We used the software *ngsLD* (v.1.1[72]) to prune our data of linked SNPs, considering that SNPs are in linkage disequilibrium (LD) if they are separated by less than 20 Kb and present a minimum weight of 0.5. PCAs were calculated with *PCAngsd*[73]. while the admixture analyses were run in *NGSAdmix*[74]. To assess convergence, we performed 50 independent runs, with the number of assumed populations (K) ranging from 1 to 5, a minimum tolerance for

convergence of $1 \times 10^{-10}$ and a minimum likelihood ratio value of $1 \times 10^{-6}$. Since we found evidence of first and second-degree relatives in our dataset, we removed one sample from each related pair (the one with lowest coverage prior to downsampling) (cf. Supplementary Table 1). *PCAngsd* and *NGSAdmix* analyses were performed on the datasets without first-degree and second-degree relatives ($n = 22$). Finally, we plotted the mean likelihood and the rate of likelihood change (or $\Delta K$[75]) across the 50 replicates for each $K$ to find the most likely $K$. Additionally, we estimated pairwise weighted Fst values for the ECU–ANT, ECU–MAG, and MAG–ANT grounds using *Vcftools* with the *--weir-fst-pop* option in the filtered genotype calls.

A haplotype median-joining network was created using the program *PopART*[76] with default settings using the whole mitogenome of the 26 HWs. In addition, we inferred a mitochondrial phylogenetic tree using W-IQ-TREE[77] with default parameters (perturbation strength 0.5 and stopping rule 100), determining the best-fit substitution model with ModelFinder ([78]TIM + F + R2), estimating branch support with 1000 ultra-fast bootstraps[79], and using the mitogenome reference of the fin whale as outgroup (MF409243.1[71]).

### Demographic history analysis
We reconstructed the demographic history of ESP-HWs surveyed in this study using the diffusion approximation method implemented in $\delta a \delta i$ (v.2.2.1[45]). First, we estimated the folded site frequency spectrum (SFS) in *Angsd* using unlinked genotype likelihoods (13,961,046 SNPs) estimated without filtering for minimum allele frequency (*-minMaf*) in the dataset including only unrelated individuals ($N = 22$). Then, following the approach of ref. 30, we fitted five nested single-population demographic models with increasing numbers of size-change epochs: 1-epoch, 2-epoch, 3-epoch, 3-epoch_TBOT5, (where the last epoch's start was fixed at 5 generations ago, roughly corresponding to 1920 when most whaling activity began) and 4-epoch models (Supplementary Fig. 9). To further evaluate the sensitivity of the 3-epoch model, we run 4 additional models where we fixed the last epoch's start at 1, 10, 20, and 30 generations ago in order to discern whether the inferred population reduction in the current 3rd epoch is related in time to the period of commercial whaling (within the last 5 generations) or to factors longer ago (like the so-called Little Ice Age (LIA) in the medieval).

We used the $\delta a \delta i$ pipeline (v.3.1.6[80]) to fit these models to our observed SFS through five rounds of optimization using the Nelder-Mead method, with [10,20,30,40,50] replicates, [3,7,10,15,20] maximum iterations per round, and [1–3]-fold perturbated starting parameters. To ensure convergence, we performed five independent pipeline runs and selected the best-fitting model based on the likelihood ratio test (LRT[81]). For models that converged, best-fit parameter sets were scaled using the equation $\theta = 4N_e \mu L$, where $L$ is the total sequence length analyzed (1,320,513,056 base pairs), $\mu$ is the mutation rate ($1.12 \times 10^{-8}$ mutations per generation per base pair[82]), and $\theta$ is the optimal theta value for the given model. We assumed a generation time of 21.5 years[44] to convert generations to years. Model uncertainty was assessed by estimating 95% confidence intervals for the best-fit parameters using 100 bootstrap pseudo-replicates generated with the *Angsd --boot* option[81].

To further validate our demographic estimations, we inferred historical changes in effective population size (Ne) over time using the software $SMC++$ (v.115.2[46]). Only unrelated individuals were included in the analysis, which was run independently for the entire population (ALL = 22), and separately for the three sampling sites (ECU = 9, MAG = 9, ANT = 4). Repetitive uncalled regions were marked as missing data (*-m*) to avoid their misidentification as very long ROH, which could compromise the power to infer demographic changes. We formed composite likelihoods by varying the distinct individual (*-d*), ran 100 iterations for each dataset (to estimate confidence intervals), assumed a generation time of 21.5 years[44] and a mutation rate of $1.12 \times 10^{-8}$[82].

We also used the LD-based method implemented in *GONE*[47] to estimate the demographic history of ESP-HWs in the recent past. We used the same genotypes and groups of individuals as in the $SMC++$ analysis and assumed a constant recombination rate of 1 cM Mb$^{-1}$, a typical

recombination rate for mammals and used in similar cetacean studies[29]. We ran *GONE* with 500 internal replicates over 500 generations, using 50,000 SNPs per chromosome and only on pairs within 2 cM ($hc = 0.02$) to reduce potential bias arising from population substructure in recent population history[29,47]. The remaining parameters were set to default, and the analysis was repeated 100 times, each time with a different randomly selected set of 50,000 SNPs per chromosome, to estimate confidence intervals.

### Genome-wide heterozygosity and ROH estimation
Genome-wide heterozygosity was estimated per sample using the genotype likelihoods by first computing the folded site allele frequency likelihood, using the reference genome as ancestral state, and then calculating the folded site frequency spectrum (SFS) with *Angsd*. Runs of homozygosity were identified on the genotype calls using the window-based approach implemented in *Plink* (v.2[83]). Sliding window size was set to 100 Kb, with a minimum of 50 SNPs, at a minimum density of 1 SNP per 50 Kb required to call a ROH and a length of 1000 Kb between two SNPs was required in order for them to be considered in two different ROHs. Finally, to account for genotyping errors, we allowed up to 3 heterozygote sites per 100 Kb window within called ROHs[54,84]. Individual inbreeding coefficients were then calculated from the extent of ROH spanning the respective reference genome (FROH = $\Sigma$ (Length of all ROH)/Total genome length; after[85]).

### Statistics and reproducibility
Analyses were based on 26 biopsy samples from free-ranging humpback whales. After removing first- and second degree relatives, $n = 22$ samples remained and were used for all subsequent analyses. These samples originated from three sites: Antarctic Peninsula ($n = 4$), Ecuador ($n = 9$), and Magellan Strait ($n = 9$). To test for potential significant differences in the estimated population genomics statistics between the three sites, ANOVAs and post-hoc Tukey tests were conducted.

### Animal research
We have complied with all relevant ethical regulations for animal research and followed the Animal Protection Law from Chile and Ecuador, respectively. Humpback whale samples in the Magellan Strait and the Antarctic peninsula were obtained by experienced researchers following bioethical guidelines of the Comité de Ética, Bioética y Bioseguridad from Universidad de Concepción (protocol number CEBB 1081–2021), Chile. Regarding sampling in the Magellan Strait, the protocol and number of samples were performed according to research permit N°E-2021-531 approved by the Subsecretaría de Pesca y Acuicultura of Chile. For Ecuador, skin samples were obtained by experienced researchers following the protocol and number of samples according to research permit of the Ministerio del Ambiente, Agua y Transición Ecológica No. 2323 of Ecuador. Moreover, sampling was performed in accordance with the local Forestry and Conservancy of Natural Protected Areas and Wildlife Law, and the guidelines of the general regulation from the Interministerial agreement no. 20140004 from Ecuador.

### Reporting summary
Further information on research design is available in the Nature Portfolio Reporting Summary linked to this article.

### Data availability
Raw read data from all individuals are available under NCBI SRA Bioproject PRJNA 1367914. All data to reproduce figures are available at Zenodo ([86]; doi: 10.5281/zenodo.18971101). See Supplementary Table 1 for sample information and Supplementary Table 6 for Heterozygosity and inbreeding statistics.

### Code availability
Bioinformatic scripts are available in a public GitHub repository under https://github.com/kikecelemin/Eastern-South-Pacific-humpback-whale-population-genomics.git. All the scripts are also available at Zenodo ([86]; https://doi.org/10.5281/zenodo.18971101).

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

### Acknowledgements
We thank Paola Acuña-Gómez, director of Fundación CEQUA, for providing the conceptual and logistical support for this study. We thank the entire team at CEQUA for their assistance with fieldwork and biopsy sampling. We are also grateful to Captain Hugo Cárdenas and the crews of the *L/M MaryPaz II* and *Huracan* for ensuring safe travel and supporting our activities in the Magellan Strait. We further thank the members of the Pacific Whale Foundation-Ecuador along with the volunteers and collaborators who contributed to sample collection in Ecuadorian waters. Special thanks go to the whale tour operator Palo Santo Travel and their team in Ecuador for their invaluable logistical support during field operations. Large-scale computations were carried out at the high-performance computing cluster managed by ZIM (Zentrum für Informationstechnologie und Medienmanagement) at the University of Potsdam. This work was supported by the Project R20F0009 "Microbiome of the external surface of keystone species of ecological and economic importance in the Magallanes region and the Chilean Antarctic: microbes as bioindicators of the aquatic ecosystem health in a global warming scenario", funded by the ANID program "Fortalecimiento del Desarrollo Científico de los Centros Regionales". Open Access funding was enabled and organized by Projekt DEAL. Field work in Ecuador was supported by funding from the Pacific Whale Foundation-Ecuador and Palo Santo Travel.

### Author contributions
E.C., J.A., L.P., and R.T. conceptualized and designed the study. L.H. performed the δaδi analysis. C.C. and J.C. provided samples and associated biological information from the Ecuador breeding ground. P.V. performed laboratory work. E.C. executed bioinformatic analyses. E.C., J.A., L.P., and R.T. interpreted the results. E.C. wrote the draft manuscript with input from J.A., L.H., L.P., P.V. and R.T. All authors edited and approved the final manuscript.

### Funding

### Competing interests
The authors declare no competing interests
