## [Transparent Peer Review file · Communications Biology]

Genomic signatures of migratory preference and historical whaling in eastern South Pacific humpback whales

Corresponding Author: Professor Ralph Tiedemann

Version 0:

Reviewer comments:

Reviewer #1

(Remarks to the Author)

This manuscript uses genomic data to assess population structure (based on both nuclear and mitochondrial sequences) and infer historical N_e from eastern South Pacific humpback whales. The work is interesting and important, and for the most part well done. However, I do have some fairly major comments with respect to the two main goals of the paper: assessing population structure and inferring demographic history. These larger issues are described in detail under my GENERAL comments. I have some much smaller comments described under the SPECIFIC heading.

GENERAL

1. POPULATION STRUCTURE. I do not think that population structure is treated appropriately here. This has implications for multiple aspects of the manuscript.

1a. Line 394: If one panmictic population ($K=1$) is a reasonable hypothesis, then it should be included in the analyses. However, the authors only test for $K=2$ and 3. Moreover, one aspect of population structure analyses is that it often uncovers unexpected patterns of structuring. Therefore, it is common for authors to include values of K higher than their general expectations. I strongly recommend that the authors re-run these analyses (using only the 22 unrelated individuals), but allow for K to range from 1-5. Then, they should report (as a table) the estimated probabilities associated with each value of K .

1b. Admixture plots: The authors conclude one panmictic population with respect to the nuclear DNA (which seems reasonable given the PCA analyses), but then they - confusingly - include figures associated with their $K=2$ admixture results in many figures (Figure 2, Figure S2, Figure S3, Figure S4, Figure S5). I think Figures S2 and S3 should be completely removed (see SPECIFIC comment #22), and the authors should remove the admixture plots from Figures S3 and S5, and Figure S4 should be removed. The Table describing the admixture results (suggested above) should be enough if $K=1$ has the highest probability. If not, then Admixture plots showing the assignment of individuals across K (from 1-5) would be helpful as a stand-alone figure.

1c. Figure S9: This plot shows data organized by "genetic cluster", but the authors are concluding that there is just one genetic cluster! These results are already presented according to each sampling location (Figure 5), and that is enough. The authors should remove Figure S9.

1d. Clarity in presentation: The authors need to pick one interpretation of the data and stick to it throughout the manuscript. Right now, they conclude one panmictic population, and divide results up by sampling location for many visualizations. This is fine, because comparisons across locations are interesting. But then they also present some results according to "genetic cluster", as described above. If there are true genetic clusters that do not coincide with sampling locations, then those should be the units of division in subsequent figures - not sampling location. As-is, with the authors using both sampling location and "genetic clusters" to view results, but then concluding that there is one genetic cluster - makes for a very unclear picture.

2. DEMOGRAPHIC HISTORY.

2a. The authors go through great lengths to describe the history of humpback whales, with an emphasis on the bottleneck

caused by industrial whaling. However, none of their dadi models (Figure S1) include a two-phase scenario of a larger ancestral population followed by a bottleneck to a smaller current N_e . I was shocked that this was not included, particularly when the authors *did* include the opposite scenario of a population expansion from a smaller ancestral N_e (without a clear justification, given the information provided in the Introduction). I strongly suggest that the authors include such a model in their analyses, and then report on it along with the others (e.g., in Figure 4A, Figure S1, Tables S2, S3, and S4). I realize that the methods used include estimates of N_e far into the past, and that whaling is very recent on such timescales, but it still seem logical to include such a scenario, particularly with the authors approach of starting with "simple" models and then getting more complex.

2b. The differences between the methods are quite large and raise concerns about any interpretation. For example, the estimates of N_e from GONE are vastly larger than those from SMC++ and dadi. Which ones should be trusted, and why? The authors should dig into this a bit more and provide the reader with more guidance. For example, can you estimate historical N_e reliably from the site-frequency-spectrum with so few samples (i.e., when they are divided by location)? I realize that this is complicated, and the answers may not be clear, but if the readers are to trust the interpretation of the authors, these differences should be explained a bit more.

SPECIFIC

1. Line 38: Change to "...baleen whale species that undertakes..."

2. Line 74: Change to "...rapidly expanded throughout the eastern South Pacific."

3. Line 81: I would recommend changing "projected" to "estimated". Projected suggests forecasting into the future, whereas the authors are describing the estimation of historical abundance.

4. Line 84: Change "...kept low abundance between 1920 and 1960..." to "remained low between 1920 and 1960"

5. Line 86: Change projected to estimated.

6. Lines 89-90: It is not clear to me what reference the authors are using to get at 181% increase compared to previous estimates. The authors should clarify the values they are using here, as well as clarify if they are referring to a 181% increase from the low estimate after whaling, or if it is 181% higher than pre-whaling estimates. This is a very important point - particularly given the context of this paper - but the math and context behind this number are not clear here.

7. Lines 91-92: It would be helpful if the authors include an estimate for the number of years and generations that this contraction lasted as context for the rest of this paragraph. As noted in the Discussion, the impact of a bottleneck on genetic diversity depends on its severity and its time relative to generation length. Therefore, if the authors have just described the effects of whaling, and are now moving onto a discussion of the potential genetic effects, then providing the reader with the context of how long the bottleneck lasted - in terms of generations - is important.

8. Line 132: Adjust references to figures as recommended in GENERAL comment 1b.

9. Lines 134-135: I recommend removing this sentence. The structuring analyses should not have included related analyses in the first place, so the results obtained when they were included are irrelevant.

10. Line 138-141: This section is nto accurate because the analyses do not allow for a full range of possibilities (e.g., $K=1$ was not included in the analyses, so individuals *had* to be divided into at least two clusters). This section should be revised after the additional analyses suggested in GENERAL comment #1a.

11. Lines 155-164: Revise after analyses including a single bottleneck are conducted, as suggested in GENERAL comment #2a.

12. Lines 191-195: Revise after revision of structuring analyses described in GENERAL comment #1a. Right now this is quite confusing.

13. Lines 204-207: I'm not sure about this interpretation. The authors should explain their logic more. Couldn't this pattern also be explained by the expansion of a successful matriline over several generations (so mutations from the "original" haplotype is observed)? The jump to the implication of a missing breeding ground seems strange to me. Perhaps it is logical, but it would need more explanation.

14. Line 212: Should change "gene flow" to "movement". The authors are talking about the migratory routes of individuals, which is a movement, not the coming together of individuals from different breeding grounds (they have already argued that it is one interbreeding population).

15. Line 243-244: Couldn't it also be the coming together of different groups (e.g., if different groups were isolated during the LGM, and then came together afterwards, wouldn't that look like expansion? Can the authors differentiate between these two scenarios?).

16. Line 373: Change analysis to analyses

17. Line 375: Change "analysis at population level" to "analyses at the population level"

18. Lines 374-377: The authors don't mention the different methods used to estimate relatedness, yet they are included in Figure 2 (i.e., KING and R1). These, and the rationale for using them, should be described in the Methods, with a justification for using both (rather than just one).

19. Figure 1: I really like Figure 1b. However, I think that it should be in the Supplementary Materials rather than in the main part of the manuscript.

20. Figure 2: Remove panel C (depending on revised admixture results)

21. Figure 3: It would be helpful if the legend stated that the labels refer to *samples* rather than names of the haplotypes. It took me a while to figure this out.

22. Figures S2 and S3: The authors should remove figures S2 and S3. These include analyses of structure including relatives. This is inappropriate and could be confusing for readers. Better to just leave it out.

23. Figure S4: Remove, depending on revised admixture results.

24. Figure S5: Remove panel B, depending on revised admixture results.

25. Figure S6: The axes should be labeled. Also, what are the lower panels showing?

26. Figure S9: Remove, depending on revised admixture results.

Reviewer #2

(Remarks to the Author)

Manuscript#: COMMSBIO-25-7292

Title: Genomic signatures of migratory preference and historical whaling on eastern South Pacific humpback whales

Authors: Enrique Celemín et al.

This manuscript offers a genomic examination of population structure and demographic history in eastern South Pacific humpback whales (ESP-HWs), drawing on whole-genome data from breeding and feeding grounds. The work addresses relevant questions regarding migratory behavior and the genetic consequences of industrial whaling. Its strengths include a comprehensive analytical strategy incorporating population structure, demographic modeling, and ROH analysis well as a nuanced interpretation of mito-nuclear discordance. The application of whole-genome data to humpback whale population genomics is timely and adds novel insights.

However, there are couple of concerns should be addressed.

General Remarks:

1. Data quality, mapping, and coverage: The reported mapping rates (consistently >99.7%) are notably high. What was the genome coverage for the sequencing? Please add this information since it is critical for the sequencing quality and reliability of the conclusions.

2. Coverage values (5–9×) are suitable for genotype likelihood-based approaches but suboptimal for confident genotype calling, demographic history inferences, or ROH detection. Please include a brief justification for downsampling to 5×. Since ROH analysis was conducted on called genotypes, low coverage increases the risk of artifactual ROH segments and underestimation of FROH. This limitation should be acknowledged in the Discussion, with appropriate citation of literature on ROH detection under low coverage.

3. The use of multiple methods (δaδi, SMC++, GONE) is a strength, but the discrepancies among them require more critical discussion. A key challenge in the manuscript is reconciling conflicting timelines of population decline across methods, as well as addressing small sample sizes for certain groups—both of which could affect the interpretation of whaling's genomic impacts. For example, for timing of the whaling bottleneck, the authors note that GONE infers a decline starting ~750 years before present (yBP), which does not align with historical records of industrial whaling (1903–1975, ~100 yBP).

4. The conclusion of panmixia is supported by PCA and ADMIXTURE. However, the analysis of gene flow could be extended beyond these conventional methods. Reporting pairwise F_{ST} values between feeding grounds (even if non-significant) would offer a quantitative measure of the low differentiation observed. Moreover, a more robust analysis of gene flow would significantly strengthen the interpretation of population connectivity, historical divergence, and potential source-sink dynamics between feeding grounds. For example, applying a method like TreeMix could help determine both the topology of population relationships and the direction and magnitude of gene flow.

Minor points:

There are some mistakes in the attached file that need to be corrected with closer attention.

Figure S5: (A) The colouring corresponds to the two main clusters found in the mitogenome haplotype network (Mito-cluster 1 in black and mito-cluster 2 in grey). The H4-9 were colored in grey, but they were assigned as Mito-cluster 1.

Figure S6: The y-axis label in the lower panels is currently missing ("[???]"). Figure S6 B-D seems like the same. Please check.

Table S3: Please report exact p-values in addition to asterisks for the likelihood ratio tests.

Reviewer #3

(Remarks to the Author)

In this paper, the authors analyzed population genomics of the humpback whales in the eastern South Pacific. This study focuses particularly on the industrial whaling period (1903 – 1975). Although genomics is a powerful tool for inferring population history, I doubt whether such a recent event is detectable by analyzing the genomes of a few individuals. Particularly, as the authors noted (L261-262), the humpback whale's longevity (~95 years) and its long generation time (~21.5 years) make this point difficult. For example, SMC-based methodologies such as SMC++ are less reliable in the recent past (<300 generations, <https://10.1093/molbev/msz191>). The LD-based methodologies, such as GONE, might be better when inferring in the recent past, though the authors prefer the results of SMC++ rather than GONE when considering demographic history in recent ages. At least, the authors should provide the age and sampling year of each sample to avoid analyzing individuals of different generations together when inferring the very recent past. My recommendation is that the authors not focus on the recent industrial whaling in this paper, or that the authors should analyze many more individuals at very young ages.

The humpback whale's genome-wide mutation rate is important in this study for estimating demographic history. The authors referred to Yim et al. (2014) about the humpback whale mutation rate ($2.77e-8$ [per site per generation], L420), but I could not find any humpback whale mutation rate values in this reference. Yim et al. (2014) estimated mutation rate of a minke whale and a fin whale as $u=1.07e-9$ [per site per year]. In addition, I do not recommend that the authors follow this reference for the mutation rate. Upon estimating the mutation rate, Yim et al. assumed that, for example, dolphins and cows diverged at 36.21 mya (Supple Table 57 of Yim et al. 2014), but this divergent time is too recent because some early fossil cetaceans are dated more than 50 mya.

The authors estimated genome-wide heterozygosity of each individual based on SFS with the assumption that the reference genome is ancestral. I think this methodology overestimates the heterozygosity because the reference genome is not ancestral and from another population (Hawaii, the North Pacific). To assess the genome-wide heterozygosity of each individual, I recommend that the authors obtain a sufficient amount of sequencing data, typically >20X.

Based on a haplotype network of the mtDNA, the authors classified the eastern South Pacific humpback whale population into two clusters (Fig. S5). I recommend that the authors divide clusters based not on a haplotype network but on monophyly on a phylogenetic tree. The "Mito-cluster 2" seems not monophyletic.

Other minor points:

L60: "show lower growth" -> "show lower annual growth rate"

L158: A period is required at the end of this sentence.

Fig. 1: Subfigures A and B are very different from each other, so these two subfigures should be divided into different figures.

Ethical concerns:

The authors collected samples using a remote biopsy system (Antarctic Peninsula) and crossbows (Ecuador and Magellan Strait). Though the authors state in the "Ethics oversight" section as "No approval/guidance required", I think approval from the relevant authorities, such as the government of Ecuador and the authors' institution(s), would be required. Please provide the collection date (at least the year) of each sample.

Version 1:

Reviewer comments:

Reviewer #2

(Remarks to the Author)

To the Editor,

I have reviewed the revised manuscript entitled "Genomic signatures of migratory preference and historical whaling in eastern South Pacific humpback whales" and the authors' point-by-point response to the reviewers' comments.

The authors have provided a satisfactory response to all substantive points raised during the initial round of review. Key revisions include the re-analysis of population structure with an appropriate range of K values (1-5), a major simplification and clarification of figures and text to present a consistent narrative of panmixia, and a significantly expanded discussion addressing the discrepancies and limitations of the demographic inference methods (δaδi, SMC++, GONE). All specific textual and graphical errors noted by the reviewers have been corrected, and ethical oversight details have been suitably added.

These revisions have substantially strengthened the manuscript. The conclusions are now logically consistent and well-supported by the presented data. The authors convincingly argue for a single panmictic population based on nuclear data while acknowledging the matrilineally-inherited structure revealed by mtDNA. Furthermore, they provide a nuanced and critical discussion of their demographic inferences, appropriately tempering claims about the precise timing of the recent decline while robustly defending the evidence for its occurrence.

I appreciate the authors' candid discussion of methodological limitations, particularly concerning the challenges of sampling a large, migratory cetacean. I agree that obtaining larger sample sizes is exceedingly difficult. However, I remain concerned that the combination of a limited number of individuals and the relatively low sequencing depth (~5X after downsampling)

imposes significant constraints on the robustness of key analyses, especially the demographic reconstructions and ROH detection. While the authors have adeptly contextualized their findings within these limitations, the inherent uncertainty introduced by the data quality suggests that the results should be interpreted with corresponding caution. For future studies of this nature, and to further solidify conclusions of this importance, investing in higher sequencing depth per individual would be highly advantageous to reduce genotype uncertainty and improve the resolution of recent demographic history.

The manuscript in its current form is methodologically sound, clearly presented, and makes a valuable contribution to the field of cetacean conservation genomics. I therefore recommend acceptance for publication in *Communications Biology*, with the understanding that the discussed data limitations are clearly acknowledged for the reader.

Optional minor suggestions for final polish:

Consider adding a brief synthesizing sentence at the end of the demographic methods discussion (lines 250-281) to more explicitly guide the reader on interpreting the combined evidence from the different temporal perspectives of each method.

The caption for Figure 4 effectively notes the different time scales. For even greater clarity, a visual marker (e.g., a shaded bar) on panels C and D indicating the historical whaling period (~1900-1975) could help readers instantly contextualize the genetic estimates.

Thank you for the opportunity to review this interesting study.

Reviewer #3

(Remarks to the Author)

The authors improved the manuscript. However, I still have some concerns about this paper.

The authors noted that “no humpback whale mutation rate is available” (L439). It is not correct. Please refer to, for example, the following paper: <https://doi.org/10.1126/science.adf2160>. Because the difference between the mutation rate estimated in this paper and that provided by Yim et al. is not subtle, re-interpretation of the results would be required (e.g., the population expansion may not be correlated with the end of LGM). In addition, because the paper mentioned above provides high-coverage WGS data of many humpback individuals, the authors can include these data in the population genomic analyses.

I still feel anxious about whether a very recent event, such as historic whaling, is detectable by the methodologies followed in this paper. For example, there used to be the Little Ice Age (LIA, 16th – 19th centuries) just before the whaling period, and it is also likely that the Ne decline is associated with the LIA. Can the authors distinguish the whaling period from the LIA based on their data?

Regarding model comparison, I recommend that the authors perform approximate Bayesian computation (ABC) to identify a very recent population decline.

Other minor comments:

L343: “buffer ATL”; this may not be a common reagent name but a reagent name provided by a specific manufacturer. So, the manufacturer’s name would be required here.

L369-371: Please provide the rate of repetitive regions removed here.

Figure S4: It might be better to reflect the number of substitutions on each branch as the branch length (e.g., the NJ tree).

Table S1: Please provide the length of called sites of each individual in this table. Furthermore, detailed information of each individual, such as sex, should also be provided.

Feedback on rebuttal to Reviewer #1:

Fig. 4: Because the Ne values estimated in Figs. 4C and 4D are profoundly different from each other, the authors should discuss the pros and cons of the SMC++ and GONE methodologies in the Discussion section to explain why such a difference occurs and which results the readers should rely on.

Version 2:

Reviewer comments:

Reviewer #3

(Remarks to the Author)

The authors have improved the manuscript. I am satisfied with the revised manuscript except for the following points (two major points and two minor points). I recommend acceptance of this manuscript if the authors address these points adequately.

L169—174: The estimated ages of the recent Ne decline do not differ from those in the previous version of the manuscript,

although the authors employed a much lower mutation rate in the revised manuscript. Please re-confirm these values.

The authors estimated that the recent N_e decline began ~750 years (30 generations) ago and became lowest around ~175 years (8 generations) ago (L169-171). These estimated ages are much older than the recent whaling period. Thus, to conclude that this N_e decline is associated with the recent whaling period, the authors must compare demographic models where the N_e sharply declined just one generation ago (probably corresponding with the recent whaling period) with models where the N_e declined between 30 and 8 generations ago, prior to the recent whaling period. I strongly recommend that the authors should employ ABC to compare between these models.

Minor points:

The authors used “yBP”, which might mean ‘years before present’. However, in chronological science, BP generally means ‘years before 1950’. Therefore, to avoid confusion, I recommend using other term such as “ya” (years ago) instead of “yBP”.

Although I mentioned the LIA as an example to suggest that the recent N_e decline is not necessarily linked to the recent whaling. Anyway, the authors replied that LIA was less severe and shorter in the Southern hemisphere. It is true, only if we consider the terrestrial habitat. However, the LIA was linked to the El Nino – Southern Oscillation (ENSO). Therefore, the offshore South American Pacific coastal area might have been severely affected by the LIA.

Rebuttal Letter:

Reviewer #1:

This manuscript uses genomic data to assess population structure (based on both nuclear and mitochondrial sequences) and infer historical N_e from eastern South Pacific humpback whales. The work is interesting and important, and for the most part well done. However, I do have some fairly major comments with respect to the two main goals of the paper: assessing population structure and inferring demographic history. These larger issues are described in detail under my GENERAL comments. I have some much smaller comments described under the SPECIFIC heading.

GENERAL

1. POPULATION STRUCTURE. I do not think that population structure is treated appropriately here. This has implications for multiple aspects of the manuscript.

1a. Line 394: If one panmictic population ($K=1$) is a reasonable hypothesis, then it should be included in the analyses. However, the authors only test for $K=2$ and 3. Moreover, one aspect of population structure analyses is that it often uncovers unexpected patterns of structuring. Therefore, it is common for authors to include values of K higher than their general expectations. I strongly recommend that the authors re-run these analyses (using only the 22 unrelated individuals), but allow for K to range from 1-5. Then, they should report (as a table) the estimated probabilities associated with each value of K .

Response: We thank the reviewer for this suggestion. We agree that including $K=1$ is important when testing for panmixia, and that exploring higher values of K can reveal unexpected population structure. We have re-run the analyses using only the 22 unrelated individuals, allowing K to range from 1 to 5. The estimated likelihoods and ΔK method for each value of K are now reported in Figure 3C and Figure S3, respectively. Although the ΔK method shows a peak at $K=3$ (Figure S3), ΔK cannot be calculated for $K=1$. Several observations support $K=1$ as the most biologically meaningful solution: (a) the highest likelihood is observed at $K=1$ (Figure 2C); (b) for $K=3$, individual assignment proportions are often evenly split (~33% each), which does not reflect realistic population structure; (c) F_{st} values among grounds are very low (Table S2); and (d) recent recommendations suggest complementing ΔK inference with visual inspection of admixture plots to avoid under or over estimating population differentiation (Stankiewicz et al., 2022). Considering the existence of two feeding grounds and the admixture proportions of the plots, $K=2$ could be considered as an alternative, however, its ΔK is lower than that of $K=3$, and the putative clusters do not

correspond to feeding or breeding grounds. Taken together, we humbly argue that our results indicate that $K=1$ provides the best representation of the eastern South Pacific humpback whales.

1b. Admixture plots: The authors conclude one panmictic population with respect to the nuclear DNA (which seems reasonable given the PCA analyses), but then they - confusingly - include figures associated with their $K=2$ admixture results in many figures (Figure 2, Figure S2, Figure S3, Figure S4, Figure S5). I think Figures S2 and S3 should be completely removed (see SPECIFIC comment #22), and the authors should remove the admixture plots from Figures S3 and S5, and Figure S4 should be removed. The Table describing the admixture results (suggested above) should be enough if $K=1$ has the highest probability. If not, then Admixture plots showing the assignment of individuals across K (from 1-5) would be helpful as a stand-alone figure.

Response: We thank the reviewer for pointing out this redundancy. We have now chosen to show the evidence of $K=1$ as Figure 2C and the admixture proportions plot in Figure S2. We agree that including separate admixture plots in multiple figures was unnecessary, so we have removed previous Figures S2, S3, and S4, as well as the admixture panels from Figures S3 and S5. This revision simplifies the presentation and should make the interpretation of population structure clearer.

1c. Figure S9: This plot shows data organized by "genetic cluster", but the authors are concluding that there is just one genetic cluster! These results are already presented according to each sampling location (Figure 5), and that is enough. The authors should remove Figure S9.

Response: We appreciate the reviewer's observation. Accordingly, we have removed Figure S9 from the supplementary material and adjusted the text to reflect this change.

1d. Clarity in presentation: The authors need to pick one interpretation of the data and stick to it throughout the manuscript. Right now, they conclude one panmictic population, and divide results up by sampling location for many visualizations. This is fine, because comparisons across locations are interesting. But then they also present some results according to "genetic cluster", as described above. If there are true genetic clusters that do not coincide with sampling locations, then those should be the units of division in subsequent figures - not sampling location. As-is, with the authors using both

sampling location and "genetic clusters" to view results, but then concluding that there is one genetic cluster - makes for a very unclear picture.

Response: We thank the reviewer for this thoughtful comment. We agree that the presentation should be consistent with our conclusion of a single panmictic population. To improve clarity, we have revised the manuscript to consistently present results as a panmictic population or by sampling location, the latter allowing for meaningful geographic comparisons without implying genetic subdivision. References to “genetic clusters” have been removed throughout the text and figures, and we have adjusted figure labels and descriptions accordingly.

2. DEMOGRAPHIC HISTORY.

2a. The authors go through great lengths to describe the history of humpback whales, with an emphasis on the bottleneck caused by industrial whaling. However, none of their dadi models (Figure S1) include a two-phase scenario of a larger ancestral population followed by a bottleneck to a smaller current N_e . I was shocked that this was not included, particularly when the authors *did* include the opposite scenario of a population expansion from a smaller ancestral N_e (without a clear justification, given the information provided in the Introduction). I strongly suggest that the authors include such a model in their analyses, and then report on it along with the others (e.g., in Figure 4A, Figure S1, Tables S2, S3, and S4). I realize that the methods used include estimates of N_e far into the past, and that whaling is very recent on such timescales, but it still seem logical to include such a scenario, particularly with the authors approach of starting with "simple" models and then getting more complex.

Response: We thank the reviewer for this comment. In fact, the scenario the reviewer is demanding was tested by us. Upon re-reading our model description, we realise the language and figures chosen to describe the model are ambiguous and potentially misleading. This potential misunderstanding centers around the definition of "epochs". While separating the past into epochs enables the model to infer population size changes among epochs, the direction of these changes (population decrease/bottleneck or increase/expansion) is not constrained, but derived from the data. Hence, the 3-epoch model that the reviewer identified as the “opposite scenario to bottleneck” is indeed a scenario which tests for a bottleneck. All our models only assume the occurrence of population size change among epochs, not its direction. Indeed, the 3-epoch model inferred a recent bottleneck likely associated with whaling (Fig. 4B). Regarding the 2-epoch model, the signal from the postglacial expansion was apparently stronger than the one from the recent bottleneck, hence the 2nd epoch was

assigned a larger population size (Fig. 4A). We have now improved the description of the models in Figure S9 and its legend to avoid possible confusion.

2b. The differences between the methods are quite large and raise concerns about any interpretation. For example, the estimates of N_e from GONE are vastly larger than those from SMC++ and $\delta a \delta i$. Which ones should be trusted, and why? The authors should dig into this a bit more and provide the reader with more guidance. For example, can you estimate historical N_e reliably from the site-frequency-spectrum with so few samples (i.e., when they are divided by location)? I realize that this is complicated, and the answers may not be clear, but if the readers are to trust the interpretation of the authors, these differences should be explained a bit more.

Response: We appreciate the reviewer's insightful comment regarding the differences among N_e estimates from GONE, SMC++, and $\delta a \delta i$. Indeed, the discrepancies highlight how each method captures different aspects of demographic history and how their assumptions influence the results. GONE infers recent changes in N_e using patterns of linkage disequilibrium (LD), making it particularly sensitive to recent demographic events, but less reliable for deeper timescales. In contrast, SMC++ and $\delta a \delta i$ estimate N_e from the site frequency spectrum (SFS) and coalescent genealogies, which provide more robust estimates over longer timescales, but require larger sample sizes and accurate SFS reconstruction. In the legend of Figure 4, we have included a caveat as to the different time scales (x-axis) among the applied approaches. When samples are subdivided by location, the reduced number of individuals per group may limit the accuracy of SFS-based methods like $\delta a \delta i$ and SMC++, leading to underestimation of N_e . Nevertheless, the SMC++ results show (1) a very similar pattern when we separated the locations (Figure S7) and when we merged them (Figure 4A) and (2) the confidence intervals we formed by running 100 iterations for each dataset are very narrow which highlights that our SMC++ analysis are robust. Likewise, our $\delta a \delta i$ models converged (Table S3) and the population and time estimates are also very narrow (Table S5), which highlights that our sample size (22 for most analysis) was enough to estimate the site-frequency spectrum reliably. To further address this reviewer's comment, we discuss further methodological aspects for all three applied methods which may cause biases in exact time and/or population size estimates. This statement reads "*GONE* requires a species-specific recombination map, which is not currently available for humpback whales. We used a recombination rate typical of large mammals (an approach previously applied in cetacean studies; Kardos et al., 2023) as well as a fixed generation time of 21.5 years, which may not fully capture life history variation. $\delta a \delta i$ and SMC++ require a mutation rate, and since the

mutation rate of humpback whales is currently unknown, we used the rate estimated for the common minke whale (Yim et al., 2014). While this does not impact the overall shape or topology of the demographic curve, it may affect the absolute timing and population size estimates which hence should not be taken literally." We further stress that – while point estimates of time and population size differ among approaches – all three approaches support a recent population decline. To this end, we shortly discuss potential other causes, but maintain our argument that commercial whaling is a likely driver of this inferred population size decrease. Notably, these inferences are inline with abundance estimates from sighting surveys, as outlined in the introduction.

SPECIFIC

1. Line 38: Change to "...baleen whale species that undertakes..."

Response: Thanks for this comment, we have addressed it directly in the manuscript.

2. Line 74: Change to "...rapidly expanded throughout the eastern South Pacific."

Response: Thanks for this comment, we have addressed it directly in the manuscript.

3. Line 81: I would recommend changing "projected" to "estimated". Projected suggests forecasting into the future, whereas the authors are describing the estimation of historical abundance.

Response: Thanks for this comment, we have addressed it directly in the manuscript.

4. Line 84: Change "...kept low abundance between 1920 and 1960..." to "remained low between 1920 and 1960"

Response: Thanks for this comment, we have addressed it directly in the manuscript.

5. Line 86: Change projected to estimated.

Response: Thanks for this comment, we have addressed it directly in the manuscript.

6. Lines 89-90: It is not clear to me what reference the authors are using to get at 181% increase compared to previous estimates. The authors should clarify the values they are using here, as well as clarify if they are referring to a 181% increase from the low estimate after whaling, or if it is 181% higher than pre-whaling estimates. This is a very important point - particularly given the context of this paper - but the math and context behind this number are not clear here.

Response: Thank you very much for pointing that out. The correct value is 18.1%, not 181%. We have revised the text accordingly and clarified that this percentage represents the increase in abundance between the 1900 and 2015 estimates. The sentence has been re-ordered to make this comparison and its context clearer.

7. Lines 91-92: It would be helpful if the authors include an estimate for the number of years and generations that this contraction lasted as context for the rest of this paragraph. As noted in the Discussion, the impact of a bottleneck on genetic diversity depends on its severity and its time relative to generation length. Therefore, if the authors have just described the effects of whaling, and are now moving onto a discussion of the potential genetic effects, then providing the reader with the context of how long the bottleneck lasted - in terms of generations - is important.

Response: Thank you very much for this suggestion. In response to the reviewer's suggestion, we have added an estimate of the duration of the bottleneck. Specifically, we now state that the bottleneck lasted approximately 70 years, which corresponds to around three HW generations. This addition provides the contextual information necessary to better interpret the potential genetic impacts of the contraction in relation to generation length.

8. Line 132: Adjust references to figures as recommended in GENERAL comment 1b.

Response: Thanks for this comment, we have addressed it directly in the manuscript.

9. Lines 134-135: I recommend removing this sentence. The structuring analyses should not have included related analyses in the first place, so the results obtained when they were included are irrelevant.

Response: We followed this suggestion and generally removed the structure analysis containing related specimens (see detailed response above).

10. Line 138-141: This section is not accurate because the analyses do not allow for a full range of possibilities (e.g., $K=1$ was not included in the analyses, so individuals *had* to be divided into at least two clusters). This section should be revised after the additional analyses suggested in GENERAL comment #1a.

Response: Thanks for this comment, we have performed the suggested additional analyses and revised this section accordingly (please see above the detailed answer to comment #1a).

11. Lines 155-164: Revise after analyses including a single bottleneck are conducted, as suggested in GENERAL comment #2a.

Response: Thanks for this comment, in fact analyses included scenarios which could infer a single bottleneck. We have now clarified this issue (see above the detailed answer to comment #2a).

12. Lines 191-195: Revise after revision of structuring analyses described in GENERAL comment #1a. Right now this is quite confusing.

Response: Thank you for this helpful comment. We believe that, after removing the respective parts of the supplementary material (as suggested above) and revising Figure 2, the presentation of the population structure results is now much clearer.

13. Lines 204-207: I'm not sure about this interpretation. The authors should explain their logic more. Couldn't this pattern also be explained by the expansion of a successful matriline over several generations (so mutations from the "original" haplotype is observed)? The jump to the implication of a missing breeding ground seems strange to me. Perhaps it is logical, but it would need more explanation.

Response: Thank you for this valuable comment. After revisiting our data and re-thinking our interpretation, we indeed agree to the reviewer that expansion of a successful matriline over several generations is a more likely scenario here, not least in light of the observation that the central haplotype of mito-cluster 1 is present in all areas, while the derived ones are mostly restricted to the Magellan Strait. We have re-formulated our interpretation as follows: "Such a pattern is commonly associated with population expansion and/or founder effects (Slatkin & Hudson, 1991; Filatova et al., 2018) and hence fits well to the postglacial expansion inferred from our nuclear analyses. Interestingly, the central haplotype of this cluster is found in all regions, while the derived haplotypes are mostly restricted to the Magellan Strait. This could indicate a particularly successful HW matriline from which HWs preferentially migrate to the Magellan Strait."

14. Line 212: Should change "gene flow" to "movement". The authors are talking about the migratory routes of individuals, which is a movement, not the coming together of individuals from different breeding grounds (they have already argued that it is one interbreeding population).

Response: Thanks for this comment, we completely agree and have addressed it directly in the manuscript.

15. Line 243-244: Couldn't it also be the coming together of different groups (e.g., if different groups were isolated during the LGM, and then came together afterwards, wouldn't that look like expansion? Can the authors differentiate between these two scenarios?).

Response: Thank you for this thoughtful comment. The (presumably post-glacial) expansion was inferred by both *δaδi* and *SMC++*, both of which inferring demography from the site-frequency spectrum (SFS). Specifically, expansion is indicated by an excess in low-frequency variants: Given a constant mutation rate, an expanding population will yield additional variants which however are rare, as they occurred only recently. Merging two populations (the scenario mentioned by the reviewer) does not yield such a pattern. Hence, we consider our inference of population expansion robust. To the best of our understanding, the SFS (as analysed in the applied methods) is uninformative as to whether there was an initial ancient population merger (although *δaδi* is capable of inferring gene flow after a population split). In other words: While our inference of population expansion is firm, our analyses are not designed to discern among scenarios of single vs. several ancient groups isolated in separate glacial refugia.

16. Line 373: Change analysis to analyses

Response: Thanks for this comment, we have addressed it directly in the manuscript.

17. Line 375: Change "analysis at population level" to "analyses at the population level"

Response: Thanks for this comment, we have addressed it directly in the manuscript.

18. Lines 374-377: The authors don't mention the different methods used to estimate relatedness, yet they are included in Figure 2 (i.e., KING and R1). These, and the rationale for using them, should be described in the Methods, with a justification for using both (rather than just one).

Response: We thank the reviewer for pointing this out. We have now added a description of the relatedness estimation methods in the Methods section. Specifically, we used both KING and R1 because they complement each other: KING is robust to population structure and can efficiently detect close kin, while R1 provides a more sensitive estimate of pairwise relatedness across the full range of relationships. Using both statistics allows us to cross-validate results and ensure that the detected relationships are robust. The rationale for

including both methods, as well as details of their implementation, are now explicitly described in the revised Methods section.

19. Figure 1: I really like Figure 1b. However, I think that it should be in the Supplementary Materials rather than in the main part of the manuscript.

Response: We thank the reviewer for this comment. We agree that subfigure B is better suited for the supplementary materials, and we have revised the manuscript accordingly.

20. Figure 2: Remove panel C (depending on revised admixture results)

Response: Thanks for this comment, we have now edited panel C to provide the likelihoods for the results K1-5.

21. Figure 3: It would be helpful if the legend stated that the labels refer to *samples* rather than names of the haplotypes. It took me a while to figure this out.

Response: Thanks for this comment, we completely agree and have addressed it directly in the manuscript.

22. Figures S2 and S3: The authors should remove figures S2 and S3. These include analyses of structure including relatives. This is inappropriate and could be confusing for readers. Better to just leave it out.

Response: We thank the reviewer for this suggestion. We agree that including analyses with related individuals could be misleading. Accordingly, we have removed Figures S2 and S3 from the supplementary material to avoid potential confusion and to ensure that the presentation focuses only on the analyses conducted with unrelated individuals.

23. Figure S4: Remove, depending on revised admixture results.

Response: We appreciate the reviewer's comment. As recommended, we have removed Figure S4 and updated Figure 2C in accordance with the revised admixture results.

24. Figure S5: Remove panel B, depending on revised admixture results.

Response: We appreciate the reviewer's comment and as stated before, we have removed all figures and writing regarding the genetic clusters.

25. Figure S6: The axes should be labeled. Also, what are the lower panels showing?

Response: Thank you for this comment, we have reorganised the subplots included labels and improved the legend.

26. Figure S9: Remove, depending on revised admixture results.

Response: We appreciate the reviewer's comment and as stated before, we have removed all figures and writing regarding the genetic clusters.

Reviewer #2:

This manuscript offers a genomic examination of population structure and demographic history in eastern South Pacific humpback whales (ESP-HWs), drawing on whole-genome data from breeding and feeding grounds. The work addresses relevant questions regarding migratory behavior and the genetic consequences of industrial whaling. Its strengths include a comprehensive analytical strategy incorporating population structure, demographic modeling, and ROH analysis well as a nuanced interpretation of mito-nuclear discordance. The application of whole-genome data to humpback whale population genomics is timely and adds novel insights. However, there are couple of concerns should be addressed.

General Remarks:

1. Data quality, mapping, and coverage: The reported mapping rates (consistently >99.7%) are notably high. What was the genome coverage for the sequencing? Please add this information since it is critical for the sequencing quality and reliability of the conclusions.

Response: Raw coverage was ~10X, please see Table S1 for the final coverage after all filtering steps.

2. Coverage values (5–9×) are suitable for genotype likelihood-based approaches but suboptimal for confident genotype calling, demographic history inferences, or ROH detection. Please include a brief justification for downsampling to 5×. Since ROH analysis was conducted on called genotypes, low coverage increases the risk of artifactual ROH segments and underestimation of FROH. This limitation should be acknowledged in the Discussion, with appropriate citation of literature on ROH detection under low coverage.

Response: We thank the reviewer for this insightful comment. We downsampled to 5× coverage to maintain consistency across samples and to allow genotype likelihood-based analyses, which are robust at lower coverage. We acknowledge that this coverage is

suboptimal for confident genotype calling, demographic history inference, and ROH detection. As suggested, we have added a discussion of this limitation, noting that low coverage can increase the risk of artifactual ROH segments and underestimation of FROH, including a reference (Ceballos et al., 2018).

3. The use of multiple methods ($\delta a\delta i$, SMC++, GONE) is a strength, but the discrepancies among them require more critical discussion. A key challenge in the manuscript is reconciling conflicting timelines of population decline across methods, as well as addressing small sample sizes for certain groups—both of which could affect the interpretation of whaling’s genomic impacts. For example, for timing of the whaling bottleneck, the authors note that GONE infers a decline starting ~750 years before present (yBP), which does not align with historical records of industrial whaling (1903–1975, ~100 yBP).

Response: We thank the reviewer for highlighting this important point. The discrepancies in the timing of population decline inferred by $\delta a\delta i$, SMC++, and GONE indeed reflect the distinct assumptions, temporal sensitivities, and data requirements of these methods. GONE infers recent N_e changes based on linkage disequilibrium (LD) decay, which can sometimes shift inferred declines further back in time if LD patterns are influenced by overlapping demographic processes or population structure predating industrial whaling. In contrast, $\delta a\delta i$ and SMC++ rely on the site frequency spectrum and coalescent genealogies, providing longer-term demographic reconstructions, but with reduced resolution for very recent events. We acknowledge that the apparent ~750 yBP decline inferred by GONE does not correspond precisely with the historical period of industrial whaling (~100 yBP). We interpret this earlier signal as potentially reflecting pre-industrial demographic fluctuations or limitations in temporal resolution due to sample size and marker density. Given the small number of individuals per regional group, we recognize that these estimates should be viewed cautiously (specially for GONE given an improbable curve for ANT). Nevertheless, we preclude to over interpret these region-specific inferences (Figure S7, S8) and we mostly focus on the dataset including all samples (Figure 4). We have revised the Discussion to explicitly compare the temporal scales and assumptions of each method and to clarify that, while GONE detects a recent decline consistent with human-driven impacts, its precise timing may be uncertain. The overall concordance across methods in detecting a population reduction remains consistent with a strong anthropogenic influence, even if the exact onset varies among models. As a similar comment was made by Reviewer 1, we also refer to our detailed answer to his/her comment 2b (see above).

4. The conclusion of panmixia is supported by PCA and ADMIXTURE. However, the analysis of gene flow could be extended beyond these conventional methods. Reporting pairwise F_{ST} values between feeding grounds (even if non-significant) would offer a quantitative measure of the low differentiation observed. Moreover, a more robust analysis of gene flow would significantly strengthen the interpretation of population connectivity, historical divergence, and potential source-sink dynamics between feeding grounds. For example, applying a method like TreeMix could help determine both the topology of population relationships and the direction and magnitude of gene flow.

Response: We thank the reviewer for these suggestions. We agree that reporting pairwise F_{ST} values between feeding grounds provides a useful quantitative measure of differentiation and we have added this analysis to the current version of the manuscript. However, we chose not to perform a TreeMix analysis because this approach requires population-level data. Given the very low F_{ST} values (Table S2) and the population structure results suggesting panmixia, we anticipated that TreeMix would likely produce spurious results rather than reflect a true biological signal.

Minor points:

There are some mistakes in the attached file that need to be corrected with closer attention.

Figure S5: (A) The colouring corresponds to the two main clusters found in the mitogenome haplotype network (Mito-cluster 1 in black and mito-cluster 2 in grey). The H4-9 were colored in grey, but they were assigned as Mito-cluster 1.

Response: We thank the reviewer for pointing this out. We have corrected the colouring in Figure S5 so that haplotypes H4–H9 are now correctly assigned to Mito-cluster 1 (black) in accordance with the mitogenome haplotype network. We have carefully reviewed the figure and legend to ensure consistency with the cluster assignments. Note that in addressing a further reviewer comment (see below), only mito-cluster 1 was inferred to be monophyletic and is retained, while the paraphyletic assemblage previously called mito-cluster 2 is now labeled "other haplotypes". Note further that this relabeling does not affect any of our inferences about non-random geographic distribution of mitochondrial lineages.

Figure S6: The y-axis label in the lower panels is currently missing (“[???”). Figure S6 B-D seems like the same. Please check.

Response: We thank the reviewer for this comment, we have reorganised the subplots, included labels and improved the legend. We included a note to address the high similarity between subplots B-D.

Table S3: Please report exact p-values in addition to asterisks for the likelihood ratio tests.

Response: We thank the reviewer for this comment, the p-values were reported in the figure legend, but we have improved the naming to make the p-values more recognisable in the figure legend.

Reviewer #3:

In this paper, the authors analyzed population genomics of the humpback whales in the eastern South Pacific. This study focuses particularly on the industrial whaling period (1903 – 1975). Although genomics is a powerful tool for inferring population history, I doubt whether such a recent event is detectable by analyzing the genomes of a few individuals. Particularly, as the authors noted (L261-262), the humpback whale's longevity (~95 years) and its long generation time (~21.5 years) make this point difficult. For example, SMC-based methodologies such as SMC++ are less reliable in the recent past (<300 generations, <https://10.1093/molbev/msz191>). The LD-based methodologies, such as GONE, might be better when inferring in the recent past, though the authors prefer the results of SMC++ rather than GONE when considering demographic history in recent ages. At least, the authors should provide the age and sampling year of each sample to avoid analyzing individuals of different generations together when inferring the very recent past. My recommendation is that the authors not focus on the recent industrial whaling in this paper, or that the authors should analyze many more individuals at very young ages.

Response: Thank you for this comment. Table S1 provides sampling year of each sample. Samples derive from the years 2010-2023 (spanning about half a HW generation). Age information is unfortunately not available. We agree to this reviewer regarding the difficulties in inferring recent demographic changes, given the long generation time and life span of the studied species and a not too large sample size. Logistically, however, we are unable to follow the suggestion to "analyze many more individuals at very young ages", as this would translate into years of additional field work. We now substantially expanded on the discussion regarding potential biases in our time estimations (see detailed responses to reviewers 1 and 2 above). We however maintain our argument of a substantial recent population decline in the

investigated HW population, as this is consistently inferred across methods, both based on LD (GONE) and the site frequency spectrum (SMC++ and dadi). Factors other than whaling may have contributed to this decline, as we shortly discuss. However, we humbly argue that relating this inferred decline to industrial whaling appears plausible, as this is in line with abundance information stemming from survey and whaling data, as outlined in the introduction. Moreover, the dadi 3-epoch model inferred a very recent population decline still affecting the current HW population. Hence, with all caution, we maintain our argument of a recent population decline related to industrial whaling.

The humpback whale's genome-wide mutation rate is important in this study for estimating demographic history. The authors referred to Yim et al. (2014) about the humpback whale mutation rate ($2.77e-8$ [per site per generation], L420), but I could not find any humpback whale mutation rate values in this reference. Yim et al. (2014) estimated mutation rate of a minke whale and a fin whale as $u=1.07e-9$ [per site per year]. In addition, I do not recommend that the authors follow this reference for the mutation rate. Upon estimating the mutation rate, Yim et al. assumed that, for example, dolphins and cows diverged at 36.21 mya (Supple Table 57 of Yim et al. 2014), but this divergent time is too recent because some early fossil cetaceans are dated more than 50 mya.

Response: We thank the reviewer for pointing this out. An incorrect inference of a deep phylogenetic split (as mentioned by the reviewer for the split between dolphins and cows) does not per se invalidate a mutation rate estimate, but could reflect incorrect (or absent) consideration of saturation, leading to a downward bias in time estimates. Nonetheless, we do agree that an estimate of the humpback whale mutation rate is critical for this study. Unfortunately, species-specific mutation rates for most cetaceans are currently not available. We therefore used the reported mutation rate of a closely related species, the minke whale (Yim et al., 2014). Similar studies on the demographic history of baleen whales have faced the same limitation and have used mutation rates from different species (e.g., Crossman et al., 2023). Notably, Nigenda-Morales et al. (2023), in a study of the demographic history of Eastern North Pacific and Gulf of California fin whales, used the same mutation rate as we applied here. Although the minke whale mutation rate may not perfectly reflect that of humpback whales, using this rate allows for direct comparison with previous studies on other baleen whales. Nevertheless, we have included a caveat in the Discussion noting this limitation and the potential uncertainty introduced by using a mutation rate from a related species.

The authors estimated genome-wide heterozygosity of each individual based on SFS with the assumption that the reference genome is ancestral. I think this methodology overestimates the heterozygosity because the reference genome is not ancestral and from another population (Hawaii, the North Pacific). To assess the genome-wide heterozygosity of each individual, I recommend that the authors obtain a sufficient amount of sequencing data, typically >20X.

Response: We thank the reviewer for this important comment. We acknowledge that using a reference genome from a different population (Hawaii, North Pacific) may introduce a slight upward bias in heterozygosity estimates. However, divergence among humpback whale populations is known to be low, and we argue that using a high-quality reference genome from another population should hence not introduce too much of a bias. Not all species (including baleen whales) have chromosome-level, high-quality reference genomes, and multitude of projects focusing on those species use the reference genome even of a different species for their analysis (Nigenda-Morales et al. (2023). The genomic footprint of whaling and isolation in fin whale populations. *Nature Communications*, 14:5465; Prasad et al. (2022) Evaluating the role of reference-genome phylogenetic distance on evolutionary inference. *Molecular Ecology Resources* 22.1 (2022): 45-55). We are fortunate that a high-quality humpback whale reference genome has recently been assembled (Carminati et al., 2024), therefore, we believe that our approach of using a high-quality reference genome from a closely related, low-divergence population is valid. Given the intermediate coverage of our data, we used genotype likelihoods for the majority of our analyses, as this approach accounts for genotype uncertainty and allows reliable SNP estimation even at low coverage (Lou et al., 2021). Similar methods for estimating genome-wide heterozygosity have been successfully applied to datasets with comparable or lower coverage (Bryc K et al. (2013) A novel approach to estimating heterozygosity from low-coverage genome sequence. *Genetics* 195(2):553-61; Celemín et al. (2025) Evolutionary history and seascape genomics of Harbour porpoises (*Phocoena phocoena*) across environmental gradients in the North Atlantic and adjacent waters. *Molecular Ecology Resources* 25, e13860; Louis et al. (2021) Selection on ancestral genetic variation fuels repeated ecotype formation in bottlenose dolphins. *Science Advances* 7, eabg1245). Nevertheless, we agree that higher coverage ($\geq 20\times$) and the use of reference genomes from the same population would provide more accurate individual-level estimates, and we have added a statement in the Discussion to acknowledge this limitation. The respective statement reads: "Furthermore, we acknowledge that our moderate coverage ($\sim 5X$) and the use of a reference genome from a different population (Hawaii, North Pacific) may

have led to an overestimation of genome-wide heterozygosity and underestimation of ROH lengths (Ceballos et al., 2018)."

Based on a haplotype network of the mtDNA, the authors classified the eastern South Pacific humpback whale population into two clusters (Fig. S5). I recommend that the authors divide clusters based not on a haplotype network but on monophyly on a phylogenetic tree. The “Mito-cluster 2” seems not monophyletic.

Response: We thank the reviewer for this comment. As suggested, we performed a phylogenetic analysis (Fig. S4), and indeed what had been labeled mito-cluster 2 by us is paraphyletic (while mito-cluster 1 is monophyletic). Consequently, we have adapted our phrasing such that we retain mito-cluster 1, while the remaining haplotypes are now referred to as "other haplotypes". Note that this rewording does not change the detected pattern of non-random distribution of mitochondrial haplotypes to breeding/feeding areas. The respective text reads: "Haplotypes present in the Magellan Strait feeding ground showed close clustering of haplotypes. This monophyletic cluster (mito-cluster 1) included the most common haplotype (H6, see Figure S5), found across all three grounds, and ten of eleven haplotypes from the Magellan Strait, which exhibited a star-like pattern with only one or two mutations separating haplotypes. The remaining haplotypes were more divergent from one another (forming several only partly resolved lineages/clusters in the phylogenetic tree, figure S4)."

Other minor points:

L60: “show lower growth” -> “show lower annual growth rate”

Response: Thanks for this comment, we have addressed it directly in the manuscript.

L158: A period is required at the end of this sentence.

Response: Thanks for this comment, we have addressed it directly in the manuscript.

Fig. 1: Subfigures A and B are very different from each other, so these two subfigures should be divided into different figures.

Response: We thank the reviewer for this helpful suggestion. We agree that subfigures A and B present distinct types of information. To improve clarity and figure organization, we have now separated them into two standalone figures, including section B now in the supplementary materials.

Ethical concerns:

The authors corrected samples using a remote biopsy system (Antarctic Peninsula) and crossbows (Ecuador and Magellan Strait). Though the authors state in the “Ethics oversight” section as “No approval/guidance required”, I think approval from the relevant authorities, such as the government of Ecuador and the authors’ institution(s), would be required. Please provide the collection date (at least the year) of each sample.

Response: Thank you very much for this comment. The data associated with each biopsy can be found in Table 1 of supplementary material, which includes, among other information, the geographical location and date of the respective collections. We further now include a new section entitled “Animal research” where we clarify the following: “We have complied with all relevant ethical regulations for animal research and followed the Animal Protection Law from Chile and Ecuador, respectively. Humpback whale samples in the Magellan Strait and the Antarctic peninsula were obtained by experienced researchers following bioethical guidelines of the Comité de Ética, Bioética y Bioseguridad from Universidad de Concepción (protocol number CEBB 1081-2021), Chile. Regarding sampling in the Magellan Strait, the protocol and number of samples were performed according to research permit N°E-2021-531 approved of by the Subsecretaría de Pesca y Acuicultura of Chile. For Ecuador, skin samples were obtained by experienced researchers following the protocol and number of samples according to research permit of the Ministerio del Ambiente, Agua y Transición Ecológica No. 2323 of Ecuador. Moreover, sampling was performed in accordance with the local Forestry and Conservancy of Natural Protected Areas and Wildlife Law, and the guidelines of the general regulation from the Interministerial agreement no. 20140004 from Ecuador.”

Rebuttal Letter:

Reviewer #2:

I appreciate the authors' candid discussion of methodological limitations, particularly concerning the challenges of sampling a large, migratory cetacean. I agree that obtaining larger sample sizes is exceedingly difficult. However, I remain concerned that the combination of a limited number of individuals and the relatively low sequencing depth (~5X after downsampling) imposes significant constraints on the robustness of key analyses, especially the demographic reconstructions and ROH detection. While the authors have adeptly contextualized their findings within these limitations, the inherent uncertainty introduced by the data quality suggests that the results should be interpreted with corresponding caution. For future studies of this nature, and to further solidify conclusions of this importance, investing in higher sequencing depth per individual would be highly advantageous to reduce genotype uncertainty and improve the resolution of recent demographic history.

Response: We thank Reviewer #2 for these thoughtful and constructive comments. We fully acknowledge that the coverage after downsampling of ~5X might represent a limitation of our study, and we appreciate the opportunity to clarify how this was addressed analytically. Importantly, our analytical strategy was specifically designed to address this potential limitation. Rather than relying on hard genotype calls, which could introduce substantial uncertainty, we based most of our downstream analyses on genotype likelihoods. The few analyses that were run on genotype calls, were on SNPs identified through genotype likelihoods. This probabilistic approach explicitly accounts for genotype uncertainty and has been shown to provide more accurate and less biased inference under low-coverage conditions. Lou et al. (2021) demonstrate that such genotype likelihood-based approaches can provide accurate results across multiple population genomic applications when working with low to intermediate coverage data, making this the methodological choice appropriate for our study. We have carefully considered the implications for our demographic reconstructions and ROH detection analyses. While we recognize that low coverage reduces resolution, particularly for very recent events and short ROH segments, we are confident that the broader demographic patterns we report (recent effective population size decline due to whaling and limited effect on humpback diversity) are robustly supported by the data. That said, we agree with the reviewer that future work would benefit from higher sequencing depth, particularly for a subset of individuals used in demographic modeling. Increasing per-individual coverage would reduce genotype uncertainty, improve the resolution of recent demographic history, and strengthen inferences about fine-scale patterns of inbreeding. We have revised the manuscript to further emphasize the uncertainty inherent in demographic inferences and to ensure that our conclusions are framed with appropriate caution. Specifically, we include the following statement in the discussion: "Future studies that incorporate temporally stratified or epigenetically aged samples, along with an annotated reference genome, and genomes sequenced at a higher coverage may help refine these estimates and provide more accurate demographic reconstructions (Wolf et al., 2022; Nigenda-Morales et al., 2023)."

Consider adding a brief synthesizing sentence at the end of the demographic methods discussion (lines 250-281) to more explicitly guide the reader on interpreting the combined evidence from the different temporal perspectives of each method.

Response: Thank you very much for this suggestion. We have now added an explicit synthesis sentence at lines 284-288 that guides readers on how to interpret the combined evidence from the complementary temporal perspectives offered by *δaδi*, SMC++, and our GONE analysis. In addition, we have included in lines 237-242 an introductory statement highlighting the strengths of each inference method. Specifically, we note that while each method operates at different historical timescales and carries distinct sources of uncertainty, the convergent pattern

across methods (the evidence for recent population decline) strengthen confidence in our interpretations.

The caption for Figure 4 effectively notes the different time scales. For even greater clarity, a visual marker (e.g., a shaded bar) on panels C and D indicating the historical whaling period (~1900-1975) could help readers instantly contextualize the genetic estimates.

Response: We appreciate this recommendation for enhanced visual clarity. However, after careful consideration, we decided not to include the shaded bars in panels C and D because, given the temporal scale of the plots, they would appear extremely close to the y-axis and thus be barely visible. We believe that adding such compressed visual elements could create clutter without meaningfully improving interpretability. Instead, we clarify the timing of the historical whaling period (~1900–1975) directly in the caption to provide appropriate context for interpreting the demographic estimates.

Reviewer #3:

The authors noted that “no humpback whale mutation rate is available” (L439). It is not correct. Please refer to, for example, the following paper: <https://doi.org/10.1126/science.adf2160>. Because the difference between the mutation rate estimated in this paper and that provided by Yim et al. is not subtle, re-interpretation of the results would be required (e.g., the population expansion may not be correlated with the end of LGM). In addition, because the paper mentioned above provides high-coverage WGS data of many humpback individuals, the authors can include these data in the population genomic analyses.

Response: We sincerely thank the reviewer for pointing out this important reference and for drawing our attention to the recently published humpback whale mutation rate. We apologize for our oversight in stating that no species-specific estimate was available. Following the reviewer’s suggestion, we repeated our demographic analyses that require a mutation rate (i.e. $\delta a\delta i$ and SMC++) using the mutation rate reported in the cited study (1.12×10^{-8}). We have updated the interpretation of the timing of demographic events in the manuscript, revised the following figures and tables: Figure 4, Figure S7 and Table S5. While the absolute timing of inferred events shifts under the new mutation rate, the overall demographic patterns and qualitative conclusions remain consistent. We also appreciate the suggestion to incorporate the high-coverage whole-genome data generated in that study. However, the individuals included there originate from the North Atlantic. Although comparing our inferences with results based on higher-coverage genomes could increase confidence in parameter estimates, such a comparison would require careful consideration of population structure and the distinct demographic history of North Atlantic populations before drawing meaningful conclusions relative to our dataset. The focus of our work is on the population structure and demographic history of eastern South Pacific humpback whales. Further, we generally agree that our inference would have been more robust by including genomes at higher coverage. However, combining genomes with different coverage into a single dataset may also introduce biases. It is indeed therefore that we downsampled those of our genomes with higher coverage to a common coverage of 5x. We consider our analytical strategy based on genotype likelihoods (see response to Reviewer #2) to be appropriate for the data and questions at hand. In addition, given the one-month revision deadline, we were not able to download, process, and harmonize these additional genomes within our analytical pipeline in a rigorous manner. We are grateful to the reviewer for highlighting this valuable resource. In particular the re-analysis with a humpback whale-specific mutation rate has improved the rigor and accuracy of our study.

I still feel anxious about whether a very recent event, such as historic whaling, is detectable by the methodologies followed in this paper. For example, there used to be the Little Ice

Age (LIA, 16th – 19th centuries) just before the whaling period, and it is also likely that the Ne decline is associated with the LIA. Can the authors distinguish the whaling period from the LIA based on their data?

Response: Thanks for this comment and the interesting idea of the Little Ice Age (LIA), rather than historic whaling, as an alternative explanation of the inferred decline of the South Pacific humpback population. Here, we have first to take into account that the timing of LIA is different among the Northern and Southern hemisphere. Specifically, it was less severe, shorter, and ended earlier in the Southern hemisphere (between about 1600 and 1700, Neukom, R., Gergis, J., Karoly, D. et al. Inter-hemispheric temperature variability over the past millennium. *Nature Clim Change* 4, 362–367 (2014). <https://doi.org/10.1038/nclimate2174>). Likely, SMC++ and GONE likely lack the precision in their time estimation to discern between historical whaling and LIA as the cause of the inferred decline. Yet, we argue that $\delta a\delta i$ supports our explanation: the 3-epoch simulation estimates the decline to happen in the current generation (generation 0), while forcing the model to implement an earlier decline results in decreased likelihoods. To further address this reviewer comment, we introduced the following statement into the discussion: "Other factors, such as climatic or anthropogenic influences may have contributed to this decline. In particular, the so-called Little Ice Age (LIA) in the medieval might have affected humpback whales. In the Southern hemisphere, LIA was less severe and shorter (between about 1600 and 1700; Neukom et al., 2014; translating into about 20 and 15 humpback generations ago). In this context, our $\delta a\delta i$ modelling argues against LIA as the major driver of the decline, as the 3-epoch simulation estimates the decline to happen in the current generation (generation 0), while forcing the model to implement an earlier decline results in decreased likelihoods (Supplementary table S3). The most plausible explanation for the recent and pronounced reduction in effective population size hence remains the well-documented period of unsustainable industrial whaling (1903–1975)."

Regarding model comparison, I recommend that the authors perform approximate Bayesian computation (ABC) to identify a very recent population decline.

Response: We appreciate Reviewer #3's suggestion to employ ABC for model comparison. Although we agree that ABC could be a useful approach for our demographic analysis, our analytical strategy already incorporates multiple complementary approaches to infer demographic history across different temporal scales. Specifically, we employed three independent software packages ($\delta a\delta i$, SMC++, and GONE) each with distinct methodological strengths and sensitivities to different historical periods. $\delta a\delta i$ excels at capturing recent and intermediate demographic changes through its reliance on the site frequency spectrum (SFS), SMC++ uses Sequential Markov Coalescent (SMC) methods to provide robust inference across longer timescales, and GONE offers direct detection of contemporary effective population size through linkage disequilibrium patterns. We have now included in lines 237-242 an introductory statement highlighting the strengths of each inference method. The convergence of evidence for a recent population decline across these three independent methodologies strongly supports our conclusions. Given this consistent signal and the differing temporal resolutions of our methods, we believe our analytical framework provides a comprehensive and well-supported demographic reconstruction. Additionally, due to the one-month revision timeline imposed by the editorial process, we prioritized repeating our demographic inference using the humpback-specific mutation rate rather than performing an ABC analysis. We are confident that this multi-method approach effectively describes both the historical and recent demographic history of eastern South Pacific humpback whales.

L343: "buffer ATL"; this may not a common reagent name but a reagent name provided by a specific manufacturer. So, the manufacturer's name would be required here.

Response: Thank you for this comment. We have now specified in line 355 the manufacturer of buffer ATL in the revised manuscript. Buffer ATL is provided by Qiagen (Hilden, Germany) as part of the DNeasy Blood & Tissue Kit.

L369-371: Please provide the rate of repetitive regions removed here.

Response: Thank you very much for this suggestion. We have now included that the humpback whale reference genome presented 1,070,425,330 bp embedded in repetitive regions (34.02 % of the genome).

Figure S4: It might be better to reflect the number of substitutions on each branch as the branch length (e.g., the NJ tree).

Response: Thank you for the suggestion. We have revised Figure S4 to reflect branch lengths proportional to the number of substitutions between haplotypes. This provides a more accurate visual representation of genetic distances among haplotypes.

Table S1: Please provide the length of called sites of each individual in this table. Furthermore, detailed information of each individual, such as sex, should also be provided.

Response: Thank you for this comment. We have revised Table S1 to include sex and the length of called SNPs for each individual. This supplementary table now provides complete individual-level documentation as requested.

Fig. 4: Because the N_e values estimated in Figs. 4C and 4D are profoundly different from each other, the authors should discuss the pros and cons of the SMC++ and GONE methodologies in the Discussion section to explain why such a difference occurs and which results the readers should rely on.

Response: Thank you very much for this comment. Following this suggestion and a similar one from Reviewer#2, we have now added an explicit synthesis sentence at the end of the demographic methods discussion (lines 284-288) that guides readers on how to interpret the combined evidence from the complementary temporal perspectives offered by *δ_adi*, SMC++, and our GONE analysis. Specifically, we note that while each method operates at different historical timescales and carries distinct sources of uncertainty, the convergent pattern across methods (the evidence for recent population decline) strengthen confidence in our interpretations.

Rebuttal Letter:

The authors have improved the manuscript. I am satisfied with the revised manuscript except for the following points (two major points and two minor points). I recommend acceptance of this manuscript if the authors address these points adequately.

Response: We thank the reviewer for the careful reading of the revised manuscript and for the constructive comments. We are pleased that the overall revisions were satisfactory, and we address each of the remaining points below.

L169—174: The estimated ages of the recent N_e decline do not differ from those in the previous version of the manuscript, although the authors employed a much lower mutation rate in the revised manuscript. Please re-confirm these values.

Response: We thank the reviewer for flagging this point and apologize for not being sufficiently explicit in our previous response. We have now re-confirmed these values: the estimated ages of the recent N_e decline did not change for $\delta a\delta i$ ($T_2 = 0$, i.e., present; Figure 4B; Table S5), while they changed slightly for SMC++ (see Figure 4C) when using the suggested mutation rate. Note that GONE does not require a mutation rate as input, so these estimates were not affected by the mutation rate change and were therefore not re-run (Figure 4D).

The authors estimated that the recent N_e decline began ~750 years (30 generations) ago and became lowest around ~175 years (8 generations) ago (L169-171). These estimated ages are much older than the recent whaling period. Thus, to conclude that this N_e decline is associated with the recent whaling period, the authors must compare demographic models where the N_e sharply declined just one generation ago (probably corresponding with the recent whaling period) with models where the N_e declined between 30 and 8 generations ago, prior to the recent whaling period. I strongly recommend that the authors should employ ABC to compare between these models.

Response: We thank the reviewer for this important suggestion. While we considered implementing ABC, this approach would have required considerable model tuning, prior specification, and summary statistic selection, a level of additional analysis that was not feasible within the revision timeline. Instead, we addressed this question by running additional 3-epoch models in $\delta a\delta i$, which we consider a robust and well-suited alternative. Compared to ABC, $\delta a\delta i$ offers direct likelihood-based inference, making model comparison more straightforward through AIC and likelihood ratio tests, without the approximations inherent to ABC. $\delta a\delta i$ also has well-established performance for demographic inference from site frequency spectra, and model comparison within this framework is computationally efficient and statistically rigorous. To specifically address this reviewer's comment, we ran additional 3-epoch models with T_2 fixed at different values ($T_2=1, 5, 10, 20,$ and 30 generations ago), including scenarios where the N_e decline occurred prior to the recent whaling period. As shown in Table S3, the 3-epoch model with the highest likelihood is the unconstrained model where T_2 was freely estimated, yielding $T_2=0$ (i.e., the present). This result confirms that the 3-epoch model best supported by our data is one where the most recent N_e decline extends to the present, consistent with an association with the recent whaling period. We adjusted the text accordingly: In methods „To further evaluate the sensitivity of the 3-epoch model, we run 4 additional models where we fixed the last epoch's start at 1, 10, 20, and 30 generations ago in order to discern whether the inferred population reduction in the current 3rd epoch is related in time to the period of commercial whaling (within the last 5 generations) or to factors longer ago (like the so-called Little Ice Age (LIA) in the medieval).“. In results: „Additionally, the 3-epoch model revealed a sharp recent decline in population size (from ~394,000 to ~2,500) in the present generation (0 generations ago) (Figure 4B). Fixing the onset of the 3rd epoch to higher values (i.e., 1, 5, 10, 20, or 30 generations) yielded lower likelihoods (Table S3).“. In discussion: „The onset of the final epoch in the 3-epoch $\delta a\delta i$ model is estimated at generation 0 (i.e., the present generation). Fixing the onset of the final epoch to higher values preceding the time of

commercial whaling (i.e., 10, 20, 30 generations ago) yields lower likelihoods (Table S3). The inferred decline in the 3rd epoch hence potentially reflects the persistence of genetic signals from the whaling-era bottleneck, which unfolded within the last five ESP-HW generations.“

The authors used “yBP”, which might mean ‘years before present’. However, in chronological science, BP generally means ‘years before 1950’. Therefore, to avoid confusion, I recommend using other term such as “ya” (years ago) instead of “yBP”.

Response: We thank the reviewer for this helpful clarification. We agree that "BP" carries a specific conventional meaning in chronological science, and that "yBP" could therefore be misleading in our context. We have replaced all instances of "yBP" with "ya" (years ago) throughout the revised manuscript.

Although I mentioned the LIA as an example to suggest that the recent Ne decline is not necessarily linked to the recent whaling. Anyway, the authors replied that LIA was less severe and shorter in the Southern hemisphere. It is true, only if we consider the terrestrial habitat. However, the LIA was linked to the El Nino – Southern Oscillation (ENSO). Therefore, the offshore South American Pacific coastal area might have been severely affected by the LIA.

Response: We thank the reviewer for this insightful clarification. We agree that our previous response was incomplete in this regard. While the terrestrial effects of the LIA were indeed less pronounced in the Southern Hemisphere, we acknowledge that the LIA was associated with intensified ENSO variability, which could have substantially affected marine productivity and prey availability along the offshore South American Pacific coast. However, our δa_i model comparison (Table S3) provides statistical support for a Ne decline extending to the present ($T_2=0$) rather than one that concluded several generations ago, as would be expected if the decline were driven predominantly by the LIA. We have revised the relevant passage such that we avoid any claim as to the severity of LIA in Southern hemisphere waters and concentrate on the affected time period. In particular in conjunction with the new results of our additional modelling of scenarios with an earlier decline (encompassing the period of LIA; see above), we note that our demographic modeling favors an association with the more recent whaling period. The rephrased statement reads: „In the Southern hemisphere, LIA was apparent between about 1600 and 1700 (Neukom et al., 2014; translating into about 20 and 15 humpback generations ago). In this context, our δa_i modelling argues against LIA as the major driver of the decline, as the 3-epoch simulation estimates the decline to happen in the current generation (generation 0), while forcing the model to implement an earlier decline results in decreased likelihoods (Supplementary table S3).“